🔓 | **Open Peer Review** | Host-Microbial Interactions | Research Article

# Oxidation of hemoproteins by *Streptococcus pneumoniae* collapses the cell cytoskeleton and disrupts mitochondrial respiration leading to the cytotoxicity of human lung cells

Anna Scasny,[1,2] Babek Alibayov,[1,2] Faidad Khan,[1,2] Shambavi J. Rao,[3] Landon Murin,[1,2] Ana G. Jop Vidal,[1,2] Perriann Smith,[4] Wei Li,[5] Kristin Edwards,[1] Kurt Warncke,[5] Jorge E. Vidal[1,2]

**ABSTRACT**  *Streptococcus pneumoniae* (Spn) causes pneumonia that kills millions through acute toxicity and invasion of the lung parenchyma. During aerobic respiration, Spn releases hydrogen peroxide (Spn-$H_2O_2$) as a by-product of the enzymes pyruvate oxidase and lactate oxidase and causes cell death with signs of both apoptosis and pyroptosis by oxidizing unknown cell targets. Hemoproteins are molecules essential for life and prone to oxidation by $H_2O_2$. We recently demonstrated that during infection-mimicking conditions, Spn-$H_2O_2$ oxidizes the hemoprotein hemoglobin (Hb), releasing toxic heme. In this study, we investigated details of the molecular mechanism(s) by which the oxidation of hemoproteins by Spn-$H_2O_2$ causes human lung cell death. Spn strains, but not $H_2O_2$-deficient Spn$\Delta spxB\Delta lctO$ strains, caused time-dependent cell cytotoxicity characterized by the rearrangement of the actin cytoskeleton, the loss of the microtubule cytoskeleton, and nuclear contraction. Disruption of the cell cytoskeleton is correlated with the presence of invasive pneumococci and an increase in intracellular reactive oxygen species. In cell culture, the oxidation of Hb or cytochrome *c* (Cyt*c*) caused DNA degradation and mitochondrial dysfunction from inhibition of complex I-driven respiration, which was cytotoxic to human alveolar cells. Oxidation of hemoproteins resulted in the creation of a radical, which was identified as a protein-derived side chain tyrosyl radical by using electron paramagnetic resonance. Thus, we demonstrate that Spn invades lung cells, releasing $H_2O_2$ that oxidizes hemoproteins, including Cyt*c*, catalyzing the formation of a tyrosyl side chain radical on Hb and causing mitochondrial disruption that ultimately leads to the collapse of the cell cytoskeleton.

**IMPORTANCE**  *Streptococcus pneumoniae* (Spn) colonizes the lungs, killing millions every year. During its metabolism, Spn produces abundant amounts of hydrogen peroxide. When produced in the lung parenchyma, Spn-hydrogen peroxide ($H_2O_2$) causes the death of lung cells, and details of the mechanism are studied here. We found that Spn-$H_2O_2$ targets intracellular proteins, resulting in the contraction of the cell cytoskeleton and disruption of mitochondrial function, ultimately contributing to cell death. Intracellular proteins targeted by Spn-$H_2O_2$ included cytochrome *c* and, surprisingly, a protein of the cell cytoskeleton, beta-tubulin. To study the details of oxidative reactions, we used, as a surrogate model, the oxidation of another hemoprotein, hemoglobin. Using the surrogate model, we specifically identified a highly reactive radical whose creation was catalyzed by Spn-$H_2O_2$. In sum, we demonstrated that the oxidation of intracellular targets by Spn-$H_2O_2$ plays an important role in the cytotoxicity caused by Spn, thus providing new targets for interventions.

**KEYWORDS**  *Streptococcus pneumoniae*, hemoproteins, oxidation, cytoskeleton, mitochondria

Address correspondence to Jorge E. Vidal, jvidal@umc.edu.

The authors declare no conflict of interest.

See the funding table on p. 23.

*Streptococcus pneumoniae* (Spn, pneumococcus) causes pneumococcal pneumonia that can progress to invasive pneumococcal disease (IPD), a leading cause of death in both children and adults (1, 2). IPD begins with the colonization of Spn in the lungs, leading to cellular death and the concurrent invasion of bacteria into the pulmonary parenchyma as well as the blood and brain, causing pneumonia, sepsis, and meningitis (1, 2). Cytotoxicity in the lung parenchyma during pneumococcal pneumonia is characterized by the inflammation of capillaries and epithelial cells, causing host cellular cytotoxicity through loss of tubulin-driven ciliary function, epithelial cell integrity, and DNA damage (3, 4). Although vaccination for children and the elderly against pneumococcal disease with a pneumococcal conjugated vaccine or pneumococcal polysaccharide vaccine has been introduced in the US, over 31,000 cases and 3,500 deaths still occur due to IPD in the nation (5).

Current literature describes that pneumolysin (Ply) and hydrogen peroxide ($H_2O_2$) produced by pneumococci have demonstrated cytotoxicity against human lung cells (1, 2, 6). $H_2O_2$ is a molecule produced by pneumococci through its glycolytic pathway (Spn-$H_2O_2$) and is produced in the highest amounts as a by-product of two enzymes, pyruvate oxidase (SpxB) and lactate oxidase (LctO). These enzymes produce $H_2O_2$ through the conversion of pyruvate to acetyl phosphate and lactate to pyruvate, respectively (7). The catalysis of pyruvate to acetyl phosphate continues with the production of acetate via acetate kinase, and ATP is formed as a by-product, representing the bulk of ATP pneumococci synthesize under aerobic conditions; therefore, the enzymatic activity of SpxB is known to contribute ~85% of the $H_2O_2$ produced by Spn, whereas LctO is responsible for the remaining ~15% of $H_2O_2$ (7, 8). Studies in our laboratory have demonstrated that under certain culture conditions, including extended incubation periods, $H_2O_2$ produced through LctO is sufficient to cause toxicity to other bacterial species such as *Staphylococcus aureus* by a mechanism(s) still under investigation (8).

Spn-$H_2O_2$ is considered an oxidant capable of oxidizing cysteine residues found on many cellular $Fe^{2+}$ metalloproteins and in cytoskeletal proteins (9, 10). During the oxidation of metalloproteins, $H_2O_2$ may react with $Fe^{2+}$, catalyzing the production of the highly reactive and toxic hydroxyl radical (˙OH), in a reaction known as the Fenton reaction [$Fe^{2+} + H_2O_2 \rightarrow Fe^{3+} + H_2O + OH^- + ˙OH$]. Products of the Fenton reaction, specifically ˙OH, are capable of oxidizing many cellular targets, including organic substrates, which can lead to adverse effects including DNA damage and ultimately cell death (11, 12). Production of ˙OH when Spn is cultured in an iron-rich medium indicated the occurrence of Fenton chemistry, and we more recently demonstrated that Spn-$H_2O_2$ oxidizes hemoglobin (Hb), releasing iron to become available for radical formation (8, 13). Under this infection-like condition, there is also the potential for the formation of a ferryl radical (˙$Hb^{4+}$) from Hb, as exogenously added $H_2O_2$ produces ˙$Hb^{4+}$ in a rate-limiting Fenton reaction (11, 12). While radical formation through iron/heme oxidation has been observed, specific radical(s) identification and role(s) in Spn cytotoxicity for lung cells have yet to be elucidated.

Spn-$H_2O_2$ causes DNA damage through double-stranded breaks, upregulation of inflammatory modulators including IL-1β and type-I interferons, and the release of apoptosis-inducing factor (4, 14–16). These biochemical processes all contribute to cell death by a mechanism that has signs of both apoptosis and pyroptosis (4, 14, 16). The cellular targets by which Spn-$H_2O_2$ triggers cell death, however, remain unknown. As aforementioned, through Fenton chemistry, $H_2O_2$ has the capability of reacting with hemoproteins (12). Hb is one such hemoprotein that makes up 95% of proteins found in red blood cells. Hb is a tetrameric protein (~64 kDa) containing two alpha and two beta subunits, and it is essential in the transport of oxygen and carbon dioxide. Each subunit contains a $Fe^{2+}$ heme center capable of reacting with $H_2O_2$, leading Hb to be considered a biologic Fenton reagent (17, 18). For example, during IPD infections, along with epithelial damage, there is an influx of red blood cells (RBCs) into alveolar spaces, providing a source of abundant red blood cells for invading pneumococci. Our

laboratory and other research teams have previously demonstrated that Spn produces a potent hemolysin (Ply) that releases Hb from RBCs (1, 19). Furthermore, our studies demonstrated that Spn-$H_2O_2$ reacts with Hb, causing its oxidation from oxy-Hb (Hb-$Fe^{2+}$) to met-Hb (Hb-$Fe^{3+}$) (19). Our team demonstrated that Spn-$H_2O_2$ and $H_2O_2$ produced by oral streptococci induce the degradation of heme from hemoglobin and the release of iron (12, 13). Clinically, the phenomenon of autooxidation of hemoglobin to ferryl Hb ($\cdot Hb^{4+}$) is observed in sickle cell disease, causing severe lung endothelial damage, cell permeability, and reduced mitochondrial function that, along with a number of other toxic direct and indirect toxic effects, ultimately leads to cell death (18, 20–22).

In the course of pneumococcal disease, besides Spn-$H_2O_2$, met-Hb (Hb-$Fe^{3+}$) can also act as an endonuclease, inducing nuclear damage (23) and potentially contributing to apoptosis (4). Moreover, hemoproteins such as hemoglobin, myoglobin, and cytochrome $c$ (Cyt$c$) have been reported to have an affinity for double-stranded DNA in the presence of $H_2O_2$ (23). Upon oxidation to their metform, these proteins are capable of cutting supercoiled plasmid DNA into circular and linear DNA (23). Cyt$c$ is a major component of the mitochondrial electron transport chain (ETC) and a key protein found in the mitochondrial intermembrane space. It functions as an electron shuttle between Complexes III and IV, the last complexes within the ETC, before ATP synthase (24). Another function of Cyt$c$ is its role in apoptosis. During programmed cell death, Cyt$c$ exits the mitochondria and is released into the cytosol to then activate a signaling cascade of caspases, which then leads to cell deconstruction (24).

Hemoproteins are involved in a plethora of essential biological processes in eukaryotic cells. These iron-containing proteins are particularly abundant (>70%) in the Golgi apparatus, endoplasmic reticulum, and endosomes, whereas hemoproteins in the cytoplasm and mitochondria represent >20% of the total protein content (25). Moreover, the oxidation of microtubules by an increase in the production of reactive oxygen species (ROS) in cardiomyocytes post-infarction has recently been linked to heart failure (10, 26). In this study, we demonstrate that Spn oxidizes hemoproteins, causing toxicity, which collapses the cytoskeleton of human lung cells and causes mitochondrial dysfunction. The discovery of this new molecular mechanism paves the way for new intervention strategies for the treatment of pneumococcal disease.

## MATERIALS AND METHODS

### Bacterial strains, media, and reagents

The *S. pneumoniae* strains utilized in this study are listed in Table 1. Strains were cultured on blood agar plates (BAP) containing 5% horse or sheep's blood from frozen stocks stored in skim milk, tryptone, glucose, and glycerin (STGG) (27) or Todd-Hewitt broth supplemented with 0.5% (wt/vol) yeast extract (THY). Reagents used and sources were the following: bovine hemoglobin (Hb-$Fe^{3+}$) (Sigma-Aldrich), catalase (Sigma-Aldrich), hydrogen peroxide (Fisher), cOmplete, Mini, EDTA-free protease inhibitor cocktail (Millipore Sigma), DNase I (Thermo-Fisher), hydroxyurea (Sigma-Aldrich), thiourea (Sigma-Aldrich), cytochrome $c$ from horse heart (Sigma-Aldrich), gentamycin (Sigma-Aldrich), glutamate (Sigma-Aldrich), malate (Sigma-Aldrich), and adenosine diphosphate (ADP) (Sigma-Aldrich).

### Preparation of complemented strains in TIGR4

We complemented *spxB* or *lctO* in TIGR4Δ*spxB*Δ*lctO* or TIGR4Δ*lctO*, respectively, as described in previous work (13). Construction of the *spxB*-complemented strain was prepared by amplifying an upstream region of the *hlpA* gene (1,540 bp), using DNA purified from TIGR4 as a template and primers, hlpA-up1 and hlp-up2, 5′TCAGCA GGTTCATGAGGGAA3′ and 5′TCATTTCTGTTTTATAACAAAGTCCGGATCCTTTAACAGCGT3′, respectively. A PCR fragment was then generated using DNA purified from TIGR4 as a template, containing the *spxB* gene (1,990 bp) and its promoter with the primers

**TABLE 1** Strains utilized in this study

| Strain | Description | Reference or source |
|---|---|---|
| TIGR4 | Invasive clinical isolate, capsular serotype 4, sensitive to antibiotics | (28) |
| TIGR4Δ*spxB* (SPJV29) | TIGR4 derivative with insertionally inactivated *spxB::ermB*, erythromycin resistance, hydrogen peroxide-deficient strain | (19) |
| TIGR4Δ*lctO* (SPJV42) | TIGR4 derivative with insertionally inactivated *lctO::aad9*, spectinomycin resistance, hydrogen peroxide-deficient strain | (8) |
| TIGR4Δ*spxB*Δ*lctO* (SPJV41) | TIGR4 derivative with *spxB* and *lctO* gene deletion by insertion-deletion with erythromycin and spectinomycin cassettes, respectively, hydrogen peroxide-deficient strain | (8) |
| TIGR4Δ*lctO*Ω*lctO* (SPJV43) | SPJV42 complemented with the gene *lctO* and its promoter | This study |
| TIGR4Δ*spxB*Δ*lctO*Ω*spxB* | SPJV41 complemented with the gene *spxB* and its promoter | This study |
| EF3030 | Invasive clinical isolate, capsular serotype 19F, sensitive to antibiotics | (29) |
| EF3030Δ*spxB* (SPJV50) | EF3030 derivative with insertionally inactivated *spxB::ermB*, erythromycin resistance, hydrogen peroxide-deficient strain | (13) |
| EF3030Δ*lctO* (SPJV51) | EF3030 derivative with insertionally inactivated *lctO::aad9*, spectinomycin resistance, hydrogen peroxide-deficient strain | (13) |
| EF3030Δ*spxB*Δ*lctO* (SPJV49) | EF3030 derivative with *spxB* and *lctO* gene deletion by insertion-deletion with erythromycin and spectinomycin cassettes, respectively, hydrogen peroxide-deficient strain | (19) |
| R6 | Laboratory unencapsulated isolate, derivative of D39, serotype 2 | (30) |
| D39 | Clinical isolate, capsular serotype 2, Avery strain | (30) |
| ATCC 33400 | Clinical isolate, capsular serotype 1 | American Type Culture Collection |

spxB1 and spxB2, 5′CTTTGTTATAAAACAGAAATGA3′ and 5′CATATTTGATTTTCGGCGAG3′, respectively. A downstream fragment containing the *cat* gene (708 bp), encoding for chloramphenicol resistance, and a fragment of open reading frame SP1114 (276 bp) were amplified using DNA from strain JWV500 as a template and primers hlpA-down1 and hlp A down 2, 5′CTCGCCGAAAATCAAATATGATCACTCACGGCATGGATGA3′ and

5′CAAAACAACATTGCCCGACG3′, respectively (2,128 bp). PCR products were purified using the QIAquick PCR purification kit (Qiagen), and the three pure PCR products were ligated (i.e., sewed) by PCR with the primers hlpA-up1 and hlpA-down2. The PCR-ligated product (5,658 bp) was verified in a 0.8% DNA gel, purified from the gel using a QIAquick gel extraction kit (Qiagen), and reamplified by PCR with primers hlpA-up1 and hlpA-down2. The PCR product was then purified as described above and used to transform TIGR4Δ*spxB*Δ*lctO* following a standard procedure (31). Transformants were harvested in BAP containing chloramphenicol (5 µg/mL) and screened by PCR using the primers spxB1 and spxB2 to confirm the complementation of *spxB*.

The *lctO*-complemented strain was constructed in a similar manner as mentioned above but using primers lcto1 (5′TGGGGTGAATAATTGGGGAAA3′) and lcto2 (5′CATTCA GTGGAGGCAATCTGT3′) to amplify a 1,387 bp product. This PCR product was purified and sewed to *hlpA* upstream and *cat*-SP1114 by PCR. The final PCR-ligated complemented construct (5,100 bp) was used to transform TIGR4Δ*lctO*, and transformants were harvested and the complementation confirmed, principally as mentioned above.

## Cell cultures of human respiratory cells

Human alveolar A549 cells (CCL-185) and human bronchial Calu-3 cells (HTB-55) were used in infection experiments. A549 cells were cultured in Dulbecco's Modified Eagle's Medium (DMEM) (Gibco) supplemented with 10% fetal bovine serum (FBS), 2 mM L-glutamine (Gibco), and 100 U/mL of penicillin-streptomycin (Gibco) in 25 cm$^2$ flasks (Fisher). Calu-3 cells were cultured in Eagle's Minimum Essential Medium (American Type Culture Collection, ATCC) supplemented with 10% FBS and 100 U/mL of penicillin-streptomycin (Gibco) in 25 cm$^2$ flasks. Cells were incubated at 37°C with 5% CO$_2$ and supplemented with fresh medium three times weekly and passaged to a new flask once weekly or when cells reached ~100% confluency.

## Preparation of inoculum

The inoculum for experiments was prepared as previously described (8, 13). Briefly, bacteria from frozen STGG stocks were plated on BAP and incubated overnight at 37°C in a 5% $CO_2$ atmosphere. Bacterial growth on BAP was then collected using phosphate-buffered saline (PBS), and this suspension contained ~$5.15 \times 10^8$ CFU/mL. Unless otherwise specified, THY or human respiratory cells were inoculated at a final density of ~$5.15 \times 10^6$ CFU/mL.

## Growth curves and bacterial density quantification

THY was inoculated with Spn strains and their mutant derivatives in a 24-well plate (Genesee Scientific) and incubated in a microplate reader (BioTek) at 37°C in a 5% $CO_2$ atmosphere. $OD_{600}$ readings were taken every 20 min for 24 h to obtain growth curve data. For experiments with bacterial density counts, THY was infected with Spn and incubated at 37°C in a 5% $CO_2$ atmosphere with thiourea, hydroxyurea, or catalase for 4 or 6 h. Following incubation, samples were taken, diluted in PBS, and plated on BAP for colony-forming units per milliliter (CFU/mL) density counts.

## Infection of human respiratory cells with *S. pneumoniae* strains

Experiments were conducted once cell cultures reached 100% confluence, which occurred within 7–10 days post-seeding. Cells were cultured in 6-well culture plates (Genesee Scientific), 6-well culture plates with glass coverslips (Genesee, Fisher), 96-well plates (Costar), or 8-well chamber slides (Lab-Tek II Chamber Slide w/ Cover, NUNC) at 37°C with a 5% $CO_2$ atmosphere. Before infection, cells were washed three times with sterile PBS (pH 7.4) and then added to the recommended cell culture medium that was supplemented with 5% FBS, 2 mM L-glutamine (Gibco) (A549 cells only), 1× HEPES (Gibco), and with or without 10 µM hemoglobin. Cells were infected with Spn or mutant derivatives and necessary controls, and infected cells were incubated for various times.

## Lactate dehydrogenase cytotoxicity assay

Experiments on the cytotoxicity of human alveolar A549 cells were performed in a 6-well plate format. Washed human alveolar cells (A549) were inoculated with Spn strains, with or without 400 or 1,000 U catalase, and incubated as mentioned above for 6 h. Following incubation, the supernatant of infected cells was collected and centrifuged for 2.5 min at $25,200 \times g$ in a refrigerated microcentrifuge (Eppendorf). Lactate dehydrogenase (LDH) release was quantified from these supernatants using the CyQuant LDH Cytotoxicity Assay Kit (Invitrogen) following the provided protocols from the manufacturer. Absorbance measurements were taken in the range of 200–800 nm using an BMG LabTech Omega spectrophotometer. The calculation of cytotoxicity follows the equation given by the kit manufacturer. For experiments with Hb, Spn strains were inoculated in THY supplemented with Hb-$Fe^{3+}$ (10 µM) and incubated for 4 or 6 h at 37°C in a 5% $CO_2$ atmosphere. After incubation, supernatants were collected, spun down for 2.5 min at $25,200 \times g$, and sterilized with a 0.22 µm syringe filter (Fisher). Sterilized supernatant (1 mL) was then incubated with confluent A549 cells in a 6-well plate and incubated for 14 h, after which LDH release from cells was quantified as mentioned above.

## Fluorescence staining of the cell cytoskeleton

To perform staining of cellular components, human alveolar A549 cells were seeded in a 6-well plate format where cell culture-treated glass coverslips were installed and used once confluent after ~8–10 days of incubation. Before the experiments, A549 cells were washed three times with PBS, added to infection medium, and infected with Spn or their mutant derivatives for 4, 6, and 8 h. After incubation with Spn, infection media was removed, and cells were washed with PBS (pH 7.4) and then fixed with 2% paraformaldehyde (PFA) overnight. Following PFA fixation, PFA was removed, and cells were washed

two times with PBS and subsequently permeabilized with 0.5% Triton X-100 in PBS for 5 min. After permeabilization, Triton X-100 was removed, and cells were washed two times with PBS and blocked with 2% bovine serum albumin (BSA) for 30 min. Tubulin was stained for 1 h with a monoclonal anti-beta tubulin antibody (Invitrogen) at a final concentration of 2 µg/mL. Cells were washed with PBS and incubated for 1 h with a secondary goat anti-mouse FITC (SouthernBiotech) at a 20 µg/mL concentration and phalloidin (7 U) in 2% BSA for 1 h. Cells were then washed with PBS, dried for 30 min, and mounted on microscope slides with ProLong gold antifade mountant containing DAPI (50 µg/mL). Slides were then imaged using a confocal Nikon A1R HD25 laser scanning confocal upright microscope.

## Quantification of cellular structures

Confocal z-stack micrographs were analyzed using the NIS-Elements Basic Research software, version 4.30.01, build 1021. For quantifying cells with microtubules, cell nuclei were quantified by moving from the bottom to the top of the z-axis in the DAPI channel for a raw count. The GFP channel corresponding to the signal of microtubules was added, and the process was repeated, taking an observation of nuclei being enclosed by microtubules as the z-axis scans through. The quantification was expressed as a simple ratio of microtubule-containing nuclei to total nuclei and converted to a percentage value. The quantification of actin cytoskeleton area and gaps between cell junctions, indicating detachment, were analyzed through the z-axis in a similar manner as the quantification of microtubules. Cell circularity was obtained by drawing a region of interest (ROI) around each cell and observing the values populated in the "Automated Measurements" window. Cell area was determined by outlining cells in an ROI using the phalloidin (red) channel and obtaining the area in $\mu m^2$. The nuclear area was performed similarly to the analysis of cellular area but by drawing an ROI around the nucleus in the DAPI channel.

## Western blotting

To quantify catalase and β-tubulin proteins, human alveolar A549 cells were seeded in a 6-well plate format and used once confluent after 10 days of incubation. This protocol follows a previously published, similar version for Western blotting (13). Before the experiments, A549 cells were washed three times with PBS, added to infection medium, and infected with Spn, a Spn mutant derivative, or 750 µM $H_2O_2$ for 16 h. After incubation, the cells were harvested, washed with ice-cold PBS, and lysed using RIPA buffer for 30 min on ice. After incubation, cells were centrifuged, samples collected, and protein concentrations were quantified using the bicinchonic acid assay (Invitrogen). The collected samples were combined with 4× reducing sample buffer, boiled, and run for 2 h at 90 V on a 12% Mini-Protean TGX precast gel (BioRad). Gels were transferred to a nitrocellulose membrane with the Trans-Blot Turbo transfer system (BioRad) and blocked for 1 h at room temperature in 5% nonfat dry milk in Tris-buffered saline supplemented with 0.1% Tween 20 (TBST). Membranes were then incubated with an anti-catalase antibody (Proteintech) at a final concentration of 1.2 µg/µL or the anti-β-tubulin antibody (Proteintech) at a final concentration of 1.6 µg/µL in 5% nonfat dry milk in TBST overnight at 4°C. Following incubation, the membrane was washed with TBST and incubated with StarBright B520 (BioRad) as the secondary antibody at 1:2,500 in 5% nonfat dry milk in TBST for 1 h at room temperature. Membranes were then washed again in TBST and then imaged on a ChemiDoc MP imager (Bio-Rad) using Image Lab 5.0 software using the StarBright B520 channel.

## Quantification of intracellular reactive oxygen species

To quantify ROS in infected cells, the CellROX Green Reagent from Invitrogen was used. Human alveolar A549 cells were seeded in an 8-well chamber slide (Lab-Tek II Chamber Slide w/ Cover, NUNC), and once confluent, cells were infected with Spn or mutant

derivatives or treated with 100 µM menadione or 750 µM $H_2O_2$ as mentioned above and incubated for 6, 8, or 16 h at 37°C in a 5% $CO_2$ atmosphere. After incubation, planktonic pneumococci were removed, and infected cells were incubated for 30 min with fresh infection medium containing the CellROX Green Reagent at a concentration of 5 µM. Cells were then washed three times with PBS and fixed with 2% PFA for 15 min, after which the cells were washed two times with PBS and stained with wheat germ agglutinin-Alexa fluorophore 555 (WGA-A555) at a concentration of 2.5 µg/mL. Cells were washed with PBS, air-dried, and mounted with ProLong gold antifade mountant containing DAPI (50 µg/mL). Cells were immediately imaged by confocal microscopy using a Nikon A1R HD25 laser-scanning confocal upright microscope. Quantification of intracellular ROS from confocal micrographs was performed using ImageJ version 1.3t (National Institutes of Health, NIH) and done by taking intensity spectrum measurements of a region of interest in the green channel (GFP) and obtaining the area under the curve. Values were expressed by creating a ratio of ROS (GFP) to DAPI intensity within the same samples.

## Hydrogen peroxide quantification

$H_2O_2$ levels were quantified in a 96-well plate format using the Amplex Red Hydrogen Peroxide Assay kit (Molecular Probes) or in the Oroboros O2k FluoRespirometer under culture conditions with THY, cell culture medium (DMEM), cell culture medium supplemented with 10 µM Hb-$Fe^{3+}$ (DMEM-Hb-$Fe^{3+}$), or with cultures of human alveolar A549 cells. Each culture condition was infected in duplicate in 6-well plates with Spn and incubated for 2, 4, or 6 h at 37°C in 5% $CO_2$. Samples quantified using the Amplex Red Hydrogen Peroxide Assay Kit follow a previously published protocol and follow the manufacturer's kit instructions for $H_2O_2$ concentration calculations (13). Samples in Fig. 4C were taken after incubation, centrifuged for 3 min at 25,200 × $g$, and $H_2O_2$ was quantified in a similar manner as above, following previously published protocols with Amplex Red but using the Oroboros O2k FluoRespirometer (32).

## Hemoglobin and cytochrome *c* oxidation assay

To analyze the oxidation of Hb, or Cyt*c*, by Spn-$H_2O_2$, THY supplemented with 10 µM Hb-$Fe^{3+}$ (THY-Hb-$Fe^{3+}$), or with 56 µM Cyt*c*, was inoculated with Spn and incubated for the indicated time at 37°C in a 5% $CO_2$ atmosphere. The supernatants were then collected, centrifuged at 4°C for 5 min at 25,200 × $g$, and transferred to a 96-well plate. In some experiments, prior to inoculation with pneumococci, THY-Hb-$Fe^{3+}$ was supplemented with DNase I [50 U], a 1× protease cocktail inhibitor, thiourea (40 or 60 mM), hydroxyurea (1, 3, 8, or 10 mM), or catalase (200, 400, 600, 800, or 1,000). The oxidation of Hb, or Cyt*c*, was determined by analyzing the spectra spanning 200 through 1,000 nm in these bacteria-free supernatants using an BMG LabTech Omega spectrophotometer.

## Detection of radicals by spin-trapping

DEPMPO [5-(diethoxyphosphoryl)-5-methyl-1-pyrroline-N-oxide, 98%, Cayman Chemical, P/N 10006435] was utilized to detect and identify the formation of radical species. Spn was cultured in THY and incubated for 4 h at 37°C. As a control, uninfected THY was incubated in the same condition. After incubation, the cell suspension was micro-centrifuged, and 32 µL of supernatant was used for each spin-trapping sample. The final reaction mixture contained, when added, 30 mM DEPMPO and 10 µM Hb-$Fe^{3+}$ in a volume of 40 µL. An Spn-$H_2O_2$-free sample was obtained by pretreating the bacterial supernatant liquid with 100 U catalase for 1 min.

The reaction mixture was immediately transferred to a Pyrex capillary tube (2 mm outer diameter), sealed at one end, and placed inside a quartz electron paramagnetic resonance (EPR) tube (4 mm outer diameter; Wilmad-LabGlass). A Bruker E500 Elex-Sys EPR spectrometer and ER4123SHQE X-band cavity resonator were used to perform continuous-wave EPR (CW-EPR) measurements under the following conditions:

microwave frequency, 9.52 GHz; microwave power, 20 mW; modulation amplitude, 0.2 mT; modulation frequency, 100 kHz; and temperature, 295 K. The spectra represent an average of 20 scans (acquisition time, 2.5 min), minus a medium (supernatant) baseline (40 scans).

Three components were deconvoluted from the simulation of the EPR spectra using the garlic algorithm in the EPR simulation software, EasySpin (33), corresponding to isotropic tumbling and the fast-motional regime in the solution. The convergence of simulations was defined by the default, local least-squares fitting criteria. An isotropic *g*-value of 2.009 was used for all samples.

## CellTox green cytotoxicity assay

As an additional assay to measure the cytotoxic response from human bronchial Calu-3 cells, the CellTox Green Cytotoxicity Assay Kit from Promega was performed using a modified version of the Endpoint Step Protocol. Calu-3 cells grown in a 96-well microplate (Costar) were washed three times with PBS and infected with Spn at a density of ~$2.75 \times 10^7$ CFU/mL. CellTox Green Reagent was added to wells, as recommended by the manufacturer, and placed in a fluorometer (BMG LabTech Flustra Omega) with incubation at 37°C. Relative fluorescence unit readings were collected every 20 min for 24 h. In some experiments, infected Calu-3 cells that had been treated with CellTox Green Reagent were imaged in the green fluorescence channel using a Nikon Eclipse TSR inverted microscope.

## Purification and quantification of extracellular DNA

Spn strain R6 was inoculated into a 6-well plate containing THY, or THY was added with 10 µM of Hb-$Fe^{3+}$. DNA purified from strain TIGR4 was added to these cultures and incubated for 4 h at 37°C under a 5% $CO_2$ atmosphere. The supernatants were then collected and centrifuged for 10 min at 12,000 × *g* in a refrigerated centrifuge (Eppendorf) and filtered using a 0.2 µm syringe filter. They were subsequently mixed with 0.5 vol of ethanol and vortexed for 10 s. The DNA from the supernatants was then purified by using the QIAamp DNA minikit (QIAGEN) following the manufacturer's instructions. To quantify amounts of extracellular DNA, a quantitative PCR (qPCR) assay targeting the *cps4A* gene was utilized (34). Reactions were performed with IQ SYBR green super mix (BioRad), with 300 nM of each primer and 2.5 µL of DNA template. Reactions were run in duplicate using a CFX96 Real-Time PCR Detection System (Bio-Rad) with the following conditions: 1 cycle at 55°C for 3 min, 1 cycle at 95°C for 2 min, and 40 cycles of 95°C for 15 s, 55°C for 1 min, and 72°C for 1 min. Melting curves were generated utilizing a cycle of 95°C for 1 min, 65°C for 1 min, and 80 cycles starting at 65°C with 0.5°C increments. For quantification purposes, standards containing $1 \times 10^3$, $1 \times 10^2$, $1 \times 10^1$, $1 \times 10^0$, $1 \times 10^{-1}$, $5 \times 10^{-2}$, or $1 \times 10^{-3}$ pg of TIGR4 DNA were run in parallel to generate a standard curve, and amounts of eDNA were calculated using the software Bio-Rad CFX manager (Hercules).

## In-gel heme detection assay

Bacteria were grown in THY with 28 µM Cyt*c* at 37°C for 2, 4, and 6 h, after which supernatants were collected, spun down, and combined with nonreducing loading buffer. The mixtures were loaded into a nondenaturing 12% Mini-Protean TGX precast gel (BioRad), which was run for 2 h at 100 V in running buffer lacking sodium dodecyl sulfate. The gel was then stained using a published protocol with o-dianisidine (13, 35).

## Animal information

Sprague-Dawley rats were purchased from Jackson Laboratory (Maine, US) at 7 weeks old and allowed to acclimate for at least 1 week before use. All animals were housed in the Center for Comparative Research animal facilities of the University of Mississippi Medical

Center (UMMC). Animals were kept on a 12-h light/12-h dark cycle and fed a standard laboratory rodent diet (Teklad 8640).

## Heart mitochondria isolation

Intact rat heart mitochondria were isolated from Sprague-Dawley rats according to published methods (36, 37). In brief, heart tissue was minced with a razor blade in 500 µL of MSM buffer (220 mM mannitol, 70 mM sucrose, 5 mM MOPS, pH 7.4) supplemented with 2 mg/mL bacterial proteinase type XXIV. The minced tissue was added to 5 mL of ice-cold isolation buffer (MSM buffer supplemented with 2 mM EDTA and 0.2% fatty acid-free BSA) and homogenized on ice with a glass homogenizer and a Teflon pestle for four strokes. To inhibit the proteinase, 0.1 mM phenylmethylsulfonyl fluoride was used. The tissue homogenate was centrifuged at $300 \times g$ for 10 min at 4°C. The supernatant was then centrifuged at $3,000 \times g$ for 10 min at 4°C, and the mitochondrial pellet was washed once in ice-cold MSM buffer. Protein concentration was determined by using the BioRad DC protein assay.

## Experiments of mitochondrial respiration

Spn strains were inoculated in a 6-well plate containing THY and incubated for 4 h at 37°C in a 5% $CO_2$ atmosphere. After incubation, supernatants were collected, spun down for 2.5 min at $25,200 \times g$, sterilized with a 0.22 µm syringe filter, and utilized for respiration testing. Intact mitochondrial respiration was measured using the Oroboros O2k FluoRespirometer at 37°C, as previously reported (36, 37). Each chamber of the O2k was calibrated for the $O_2$ concentration in nano-pured water. Complex I-driven respiration was measured using isolated heart mitochondria in a buffer containing 10 mM KPi, 5 mM $MgCl_2$, 30 mM KCl, 1 mM EDTA, and 75 mM Tris, pH 7.5, with 100 µL of the Spn supernatants. The total volume of the assay mixture was 2.1 mL. State 2 respiration was initiated by adding 20 mM glutamate and 10 mM malate. State 3 respiration (ATP synthesis) is achieved by adding 2 mM ADP.

## RESULTS

### *S. pneumoniae*-produced $H_2O_2$ is a main contributor to cytotoxicity against human alveolar cells

To gain insights into the molecular mechanism of Spn-$H_2O_2$ causing cytotoxicity in pulmonary cells, we investigated the cytotoxic effect of pneumococcal strains that produce different amounts of this pro-oxidant against human alveolar cells. For this purpose, we used TIGR4 that produces ~700 µM of $H_2O_2$ within 6 h of incubation or mutant derivatives TIGR4Δ*spxB* and TIGR4Δ*lctO* that are defective for $H_2O_2$ production (8, 13). Despite SpxB and LctO being the main producers of ATP, the growth of Spn strains mutated in either enzyme (Δ*spxB*Δ*lctO*) shows no deficiency in growth (Fig. 1A). We demonstrated that Spn-$H_2O_2$ produced by TIGR4Δ*spxB* [i.e., through enzyme LctO] during an ~24-h incubation period was lethal for other bacterial species when incubated along with Spn, thereby investigating the cytotoxicity of a double mutant, TIGR4Δ*spxB*Δ*lctO*, that does not produce $H_2O_2$ (8).

Cytotoxicity was evaluated using the CyQuant LDH Cytotoxicity Assay (LDH assay). Compared with the positive control that corresponds to 100% of cytotoxicity, infection of human alveolar A549 cells with TIGR4 wild type (WT) or TIGR4Δ*lctO* caused 26.8% or 25.3% cytotoxicity 6 h post-inoculation, respectively (Fig. 1B). In contrast, infection of alveolar cells with TIGR4Δ*spxB*Δ*lctO* or TIGR4Δ*spxB* caused a significant decrease in the level of cytotoxicity (Fig. 1B).

In strain TIGR4, the activity of SpxB is also linked to the biosynthesis of the capsule polysaccharide (38). To account for other potential pleotropic effects due to a mutation in *spxB*, human alveolar cells were inoculated with TIGR4, and $H_2O_2$ was scavenged with catalase. Remarkably, in control A549 cells treated with 400 or 1,000 U of catalase, 10.6% or 20.8% cytotoxicity was observed, respectively (Fig. 1C). Cytotoxicity was, however,

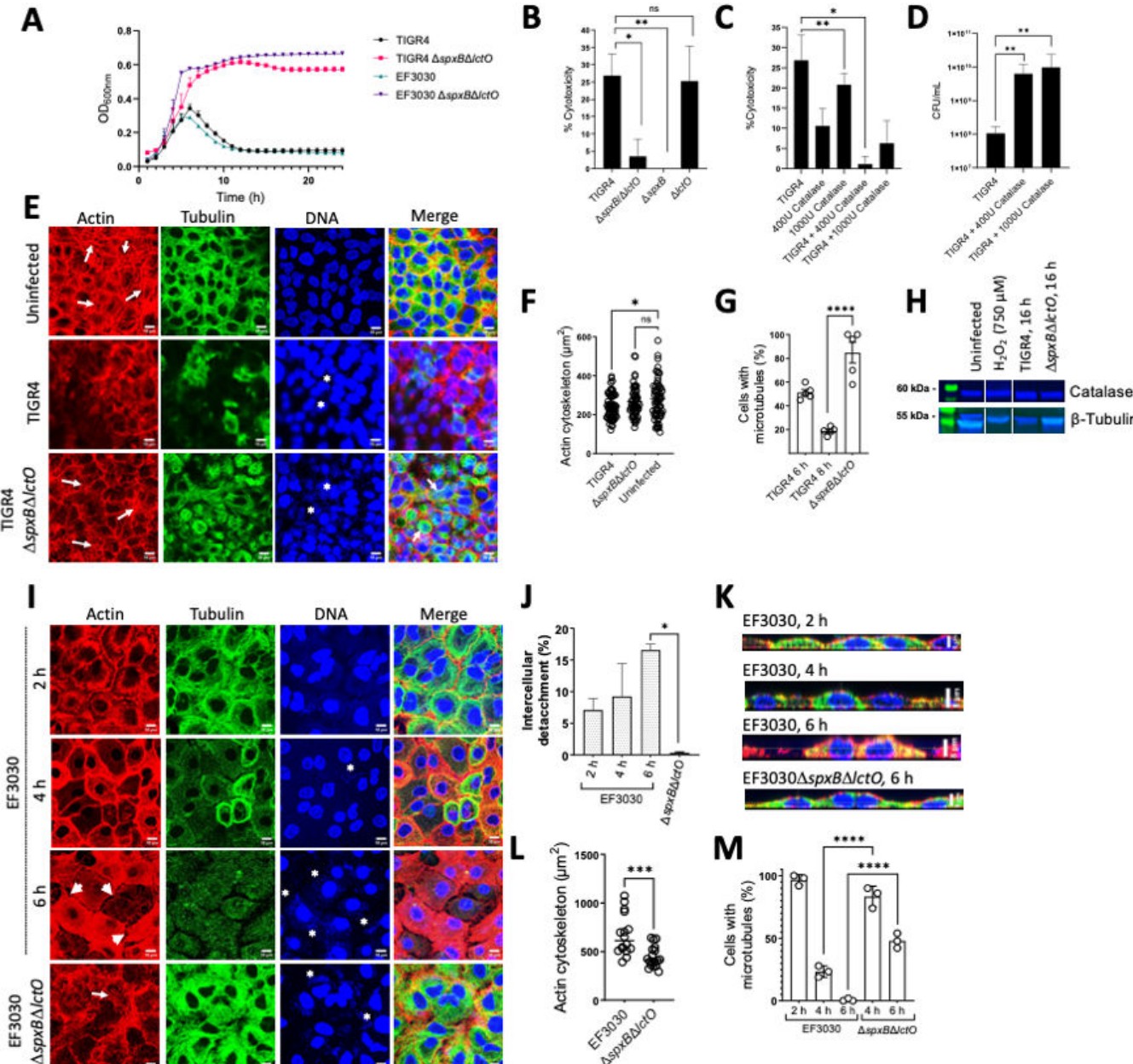

**FIG 1** Cytotoxicity by Spn-$H_2O_2$ caused actin rearrangement and the loss of microtubules. (A) Growth curves at $OD_{600}$ of TIGR4, EF3030, and their mutant derivates when grown in THY for 24 h. (B) Human alveolar A549 cells were infected with TIGR4, or mutant derivatives, and were incubated for 6 h. (C) Another group of A549 cells were treated with catalase alone, infected with TIGR4, and treated with catalase or infected with TIGR4. Cells were incubated as above. Supernatants from B to C were collected, and cell cytotoxicity was assessed using the LDH assay. Cytotoxicity percentages were calculated following the manufacturer's protocols. (D) THY containing catalase was infected with TIGR4 for 6 h, and following incubation, bacteria were collected and plated to obtain CFU/mL density. (E) Confocal micrographs of human alveolar A549 cells in DMEM infected with TIGR4 or its mutant derivative grown in 6-well plates with glass coverslips for 8 h. Actin was treated with phalloidin (red), tubulin was stained with anti-beta tubulin (green), and DNA was stained with DAPI (blue). Z-stacks of 0.3 μm each from ~10 μm height and XY optical middle sections are shown in all panels. As a control, A549 cells were left uninfected but under the same culture and staining conditions as infected cells. Asterisks indicate TIGR4 presence, and arrows indicate stress fibers. (F-G) Representative quantification of confocal micrographs from E for actin cytoskeleton area in μm² (F) and cells with microtubules present expressed as a percentage ratio of cells containing microtubules to total cells (G). (H) Representative Western blot of human alveolar A549 cells that were uninfected, infected with TIGR4, mutant derivatives, or 750 μM $H_2O_2$ and incubated for 16 h. Supernatants were collected, run on a 12% polyacrylamide gel, and stained for catalase and β-tubulin. (I) Confocal micrographs of human alveolar A549 cells in DMEM infected with EF3030, as performed in E. Asterisks indicate EF3030 presence, and arrows indicate stress fibers. (J, L-M) Representative quantification of confocal micrographs from I for (J) intercellular spaces (i.e., gaps) (L) actin cytoskeleton area in μm² (M) percentage of cells with a microtubule

**FIG 1** (Continued)

cytoskeleton. (K) Representative xz cross-sections of confocal micrographs from I. Error bars represent the standard errors of the means calculated using data from at least two independent experiments. The level of significance was determined using a *t*-test. *, $P < 0.05$; **, $P < 0.01$; ***, $P < 0.001$; ****, $P < 0.0001$; ns, non-significant. (A) Created with BioRender.

inhibited in A549 cells infected with TIGR4 and treated with 400 U catalase, whereas those infected with TIGR4 and treated with 1,000 U catalase showed residual 6.2% cytotoxicity, perhaps due to the excess of catalase. To account for the possible similar cytotoxic effects catalase may have on Spn, Spn was cultured alongside 400 or 1,000 U catalase, and CFU/mL counts were performed. These counts found that catalase is not cytotoxic to Spn but aids in the protection of pneumococci from self-produced $H_2O_2$ (Fig. 1D). Therefore, Spn-$H_2O_2$ is a main contributor to the cytotoxicity of human alveolar cells.

## Infection of alveolar cells with *S. pneumoniae* collapses the cell microtubules and actin cytoskeleton via intracellular release of $H_2O_2$

The contraction of the cell cytoskeleton is a hallmark of those undergoing cell death, and increased oxidative stress leads to remodeling of the microtubule cytoskeleton, causing dysfunction of the pulmonary endothelial cell barrier and dysfunction of cardiomyocytes (10, 26, 39). We therefore stained the actin cytoskeleton and the microtubule cytoskeleton of human alveolar cells infected with Spn strain TIGR4 (Fig. 1E) or EF3030 (Fig. 1I). Uninfected human alveolar A549 cells grown to confluency for ~10 days as well as A549 cells infected with Δ*spxB*Δ*lctO* showed abundant stress fibers (Fig. 1E and I, arrows). The area of the actin cytoskeleton in uninfected alveolar cells or cells infected with TIGR4 Δ*spxB*Δ*lctO* was similar, with a median of 265 and 253 µm², respectively (Fig. 1F). In contrast, the actin cytoskeleton of human alveolar cells infected with TIGR4 underwent reorganization and was characterized by the loss of stress fibers (Fig. 1E). These TIGR4-infected cells also had a significantly contracted actin cytoskeleton area (median, 231 µm²) compared to uninfected cells (Fig. 1F). The actin cytoskeleton of cells infected with EF3030 showed cell swelling and detachment at the cell junctions (Fig. 1I, arrowheads, and Fig. 1J). Unlike TIGR4 infection, an infection of human alveolar cells with strain EF3030 caused an increased actin cytoskeleton area, as evaluated by confocal xz (Fig. 1K) and yz optical sections (not shown) and quantified from confocal micrographs (Fig. 1L).

In these confocal optical sections, uninfected A549 cells showed a well-defined microtubule network (Fig. 1E). The microtubule cytoskeleton of TIGR4-infected and EF3030-infected alveolar cells was gradually lost after this 8-h incubation period (Fig. 1E and I). We then quantified alveolar cells containing an intact microtubule cytoskeleton to demonstrate a time-dependent loss of the microtubule cytoskeleton in alveolar cells infected with TIGR4 (Fig. 1) or EF3030 (Fig. 1M), whereas the microtubule cytoskeleton was intact in 84.8% or 45.1% of alveolar cells infected with TIGR4Δ*spxB*Δ*lctO* or EF3030 Δ*spxB*Δ*lctO*, respectively (Fig. 1G and M). Western blot analysis of alveolar cells infected with pneumococci revealed that the microtubules were degraded in cells infected with TIGR4, but the signal of the microtubules remained intact in those infected with TIGR4Δ*spxB*Δ*lctO* (Fig. 1H). Signal of microtubules with an addition of 750 µM $H_2O_2$ highlights the continuous production of $H_2O_2$ over time by TIGR4 to induce significant damage to cells. The signal of catalase, however, was observed in all alveolar cells regardless of whether they were infected with $H_2O_2$-producing TIGR4, further indicating that the loss of the microtubule network and its degradation are triggered by oxidative reactions.

To further investigate if the absence of cytoskeletal damage was not due to a defect of the Δ*spxB*Δ*lctO* mutants to invade lung cells, we analyzed the eukaryotic nuclei and bacterial DNA in the same preparations that were stained with DAPI. Confocal bottom-middle optical sections revealed that TIGR4, TIGR4Δ*spxB*Δ*lctO*, EF3030, or EF3030Δ*spxB*Δ*lctO* pneumococci had invaded alveolar cells, indicating that cytoskeleton toxicity was due to Spn-$H_2O_2$ (Fig. 1E and I, asterisks).

Spn-$H_2O_2$ can be secreted in the supernatant and diffuse inside the cells, or it can be directly released in the cell cytoplasm by intracellular pneumococci (Fig. 2A); regardless, Spn-$H_2O_2$ may increase intracellular ROS, which increases toxicity. We therefore quantified the oxidative stress inside the cells by modeling ROS activity in cultured human alveolar A549 cells infected with pneumococci. ROS production was quantified using CellROX Green from the middle sections (i.e., intracellular) of z-stack confocal micrographs. Compared to uninfected alveolar cells that showed basal levels of ROS, A549 cells infected with TIGR4, EF3030, or menadione (100 µM) used as a positive control (40) showed a significant increase of ROS (Fig. 2B through D). In contrast, the level of ROS in cells infected with TIGR4Δ*spxB*Δ*lctO* or EF3030Δ*spxB*Δ*lctO* was similar to that in uninfected cells (Fig. 2D). In A549 cells that were treated with 750 µM $H_2O_2$, a minimal increase in ROS is seen but is significantly less than TIGR4 or EF3030 (Fig. 2B and C). This, again, is in part due to Spn's continuous production of $H_2O_2$ and its capability to increase ROS in cells over extended periods. We also stained the membrane of alveolar cells with wheat germ agglutinin, and confocal micrographs obtained in the xz and yz focal planes confirmed an increase in intracellular ROS in those infected with pneumococci producing hydrogen peroxide compared with Δ*spxB*Δ*lctO* strains (Fig. 2C, enlargement). Intracellular pneumococci were observed in the middle sections of micrographs stained with DAPI (Fig. 2C). Altogether, these results indicate that the cytotoxicity induced by Spn-$H_2O_2$ causes an increase in intracellular ROS and the collapse of the cell's actin and microtubule cytoskeleton, leading to nuclear contraction.

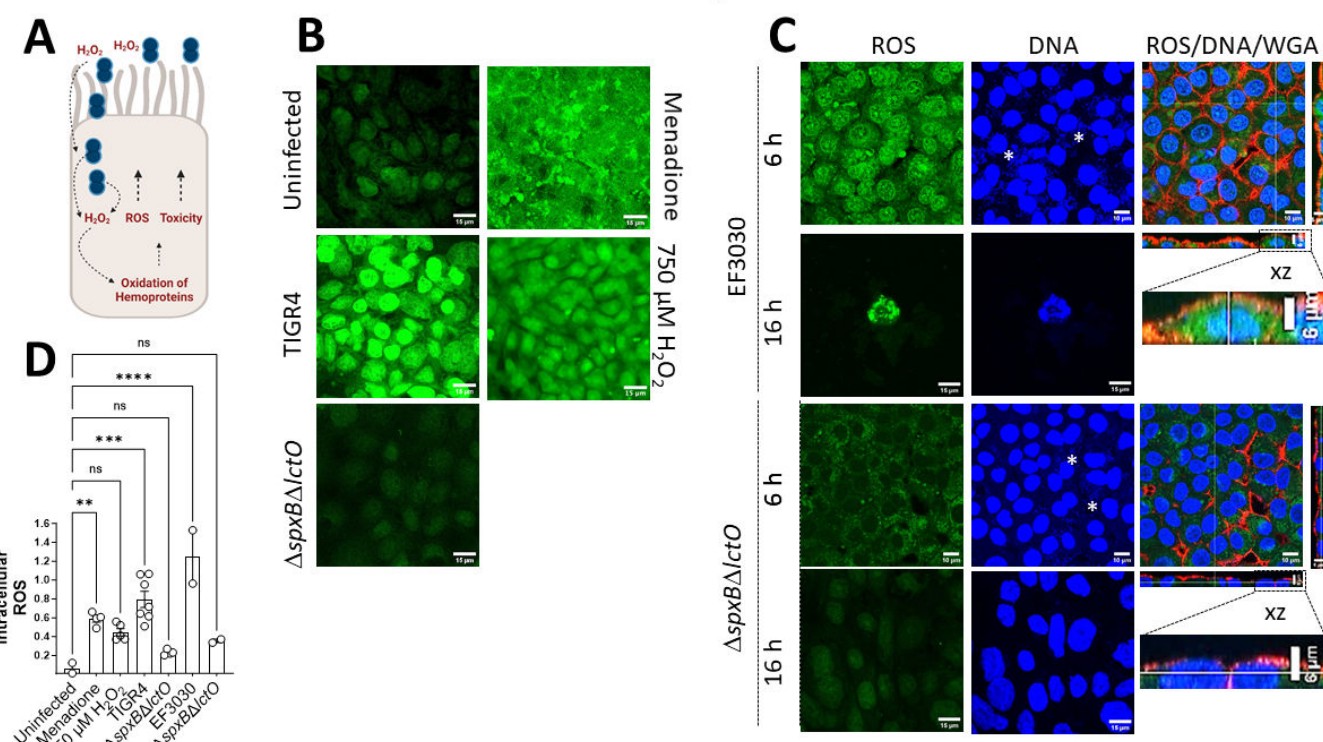

**FIG 2** Increased ROS in cells infected with Spn-$H_2O_2$. (A) Schematic representation of an increase in ROS leading to toxicity in human alveolar cells. Extracellular and intracellular pneumococci produce $H_2O_2$, which increases the pool of ROS but also oxidizes hemoproteins, further increasing other ROS molecules, leading to cell toxicity. (B-C) Human alveolar A549 cells were infected with TIGR4, TIGR4Δ*spxB*Δ*lctO*, EF3030, EF3030Δ*spxB*Δ*lctO*, treated with 100 µM menadione or 750 µM $H_2O_2$, or left uninfected and incubated for (B) 8, (C) 6, or 16 h. Post-incubation, cells were treated with CellROX (5 µM), fixed, and stained with DAPI (DNA); the membranes of some cells were also stained with WGA-Alexa 555. Confocal optical XY middle sections are shown except the far-right panel, which also shows XZ and YZ optical sections. (D) Intracellular ROS was quantified from confocal micrographs as detailed in Materials and Methods and reported as a ratio of GFP:DAPI channel intensity within the same samples. Error bars represent the standard errors of the means calculated using data from at least two independent experiments. The level of significance was determined using a one-way analysis of variance with Dunnett's post hoc *t*-test. **, $P < 0.01$; ***, $P < 0.001$; ****, $P < 0.0001$; ns, non-significant.

## Spn-$H_2O_2$ oxidation of the hemoprotein Hb-Fe$^{3+}$ induces cytotoxicity in human lung cells

Along with the cell cytoskeleton, the molecular target(s) of Spn-$H_2O_2$ in human cells have not yet been described. Other intracellular targets prone to oxidation include hemoproteins, which are abundant in eukaryotic cells and vital for cellular processes (25). We recently demonstrated that Spn-$H_2O_2$ oxidizes Hb-Fe$^{2+}$, releasing free iron, leading to the possibility of Fenton chemistry to produce ˙OH (13, 19, 41). Because of the complexity of studying the oxidation of hemoproteins inside the cells, as a surrogate, we assessed whether the oxidation of Hb contributes to the cytotoxicity of human lung cells. To test this hypothesis, TIGR4 or TIGR4$\Delta spxB\Delta lctO$ was grown in THY containing 10 µM of Hb-Fe$^{3+}$ for 6 h. We confirmed that oxidation of Hb-Fe$^{3+}$ occurred in supernatants harvested from cultures of TIGR4 but not in those obtained from TIGR4$\Delta spxB\Delta lctO$ (not shown). These supernatants were then collected, filter-sterilized, and added to cultures of A549 cells, and treated cells were incubated for 14 h. TIGR4 supernatants induced a significant, 10-fold, increased cytotoxicity of A549 cells relative to supernatants from TIGR4$\Delta spxB\Delta lctO$ (Fig. 3A). Given that Hb-Fe$^{3+}$ has pseudoperoxidase activity and its oxidation releases iron (13), we hypothesized that Spn-$H_2O_2$ oxidation of Hb-Fe$^{3+}$ catalyzes the production of toxins and radicals to cause this observed cytotoxicity (Fig. 3B).

To investigate the molecular basis of the above hypothesis, human alveolar cells were infected with Spn with or without Hb-Fe$^{3+}$, and $H_2O_2$ was quantified in the supernatants. We analyzed culture supernatants of THY, THY supplemented with Hb-Fe$^{3+}$, DMEM, and DMEM supplemented with Hb-Fe$^{3+}$ that were inoculated with Spn and incubated at 37°C. In addition, we inoculated human alveolar A549 cell cultures in which the cell culture medium (DMEM) was supplemented with or without Hb-Fe$^{3+}$ (Fig. 3C), and $H_2O_2$ was quantified in these supernatants. As hypothesized, there was a statistically significant, time-dependent, increase in $H_2O_2$ detection in the supernatants when pneumococci grew in DMEM for 2 (3.3 µM), 4 (58.9 µM), or 6 h (643.2 µM) (Fig. 3C). Detection of $H_2O_2$, however, significantly dropped to 110 µM when Spn was inoculated in DMEM + Hb-Fe$^{3+}$ and incubated for 6 h, suggesting that the pseudoperoxidase activity of Hb-Fe$^{3+}$ had catalyzed the conversion of $H_2O_2$ to water and $O_2$, and therefore it was undetectable through the assay (Fig. 3C). Spectroscopic analysis of DMEM-Hb-Fe$^{3+}$ culture supernatants infected with TIGR4 confirmed further oxidation leading to heme release and/or degradation at 2, 4, and 6 h post-inoculation, compared with the uninfected control (Fig. 3D).

A decreased detection of $H_2O_2$ was also observed in THY supplemented with Hb-Fe$^{3+}$ and incubated for 2, 4, or 6 h compared to THY lacking Hb-Fe$^{3+}$ (Fig. 3E), supporting that the cytotoxicity observed in Fig. 3A was in part due to toxins catalyzed by the oxidation of Hb-Fe$^{3+}$. Interestingly, when human alveolar A549 cells were infected with Spn in DMEM with no Hb-Fe$^{3+}$, levels of $H_2O_2$ significantly decreased at 2, 4, and 6 h post-infection, indicating that this pro-oxidant had reacted with its cell target(s) (Fig. 3C). Collectively, these results indicate that Spn-$H_2O_2$ produced under these culture conditions fully reacts with its target(s) in human cells to cause damage, in part through the creation of toxic radicals (Fig. 3B).

## Identification of a potential Hb-Fe$^{3+}$-derived molecule created by a reaction with Spn-$H_2O_2$

Previous studies have demonstrated that the oxidation of Hb generates heme degradation products that can be detected by spectroscopy (42). Remarkably, our spectroscopic studies identified a time-dependent rise in absorbance at wavelengths below the Soret band that extended into the ultraviolet (UV) region below our detection limit of 300 nm that occurred during the oxidation of Hb-Fe$^{3+}$ by Spn-$H_2O_2$. This rise in absorbance, characterized by a shoulder at ~305 nm and hereafter denoted as the 305 nm species, was produced by Spn when incubated with Hb-Fe$^{3+}$ for up to 6 h (Fig. 4A). There was a

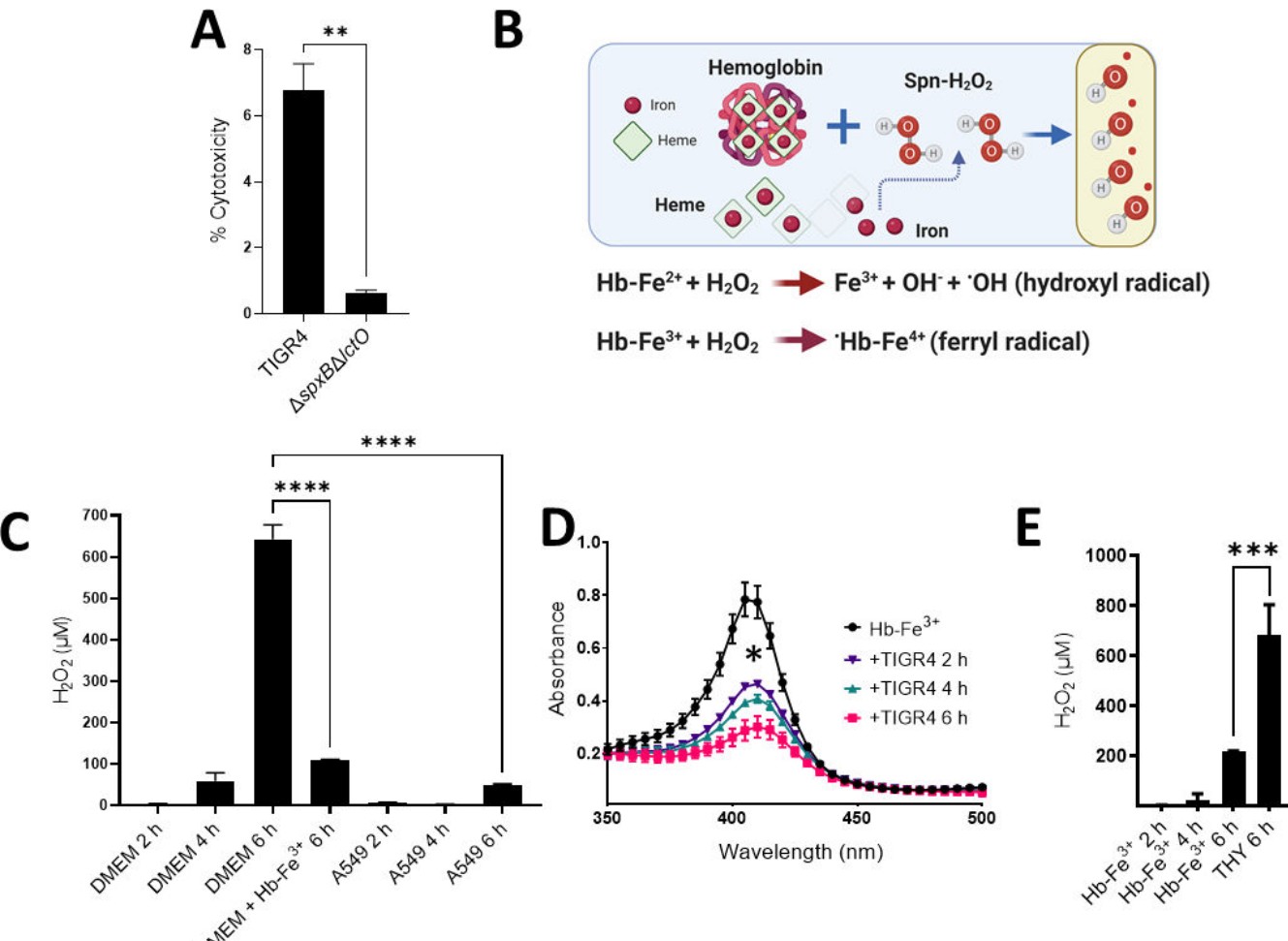

**FIG 3** Cytotoxic role and peroxidase activity of Hb-Fe$^{3+}$ with Spn-H$_2$O$_2$ and human cells. (A) Human alveolar A549 cells were incubated for 14 h with sterilized TIGR4 supernatants grown for 6 h in THY containing Hb-Fe$^{3+}$. (B) Proposed mechanisms for radical production by Spn-H$_2$O$_2$ through the Fenton reaction. Spn scavenges free iron through Hb oxidation, and Spn-H$_2$O$_2$ reacts with the free iron to produce ˙OH or Hb-Fe$^{4+}$. (C) Cell culture medium (DMEM), DMEM supplemented with 10 µM Hb-Fe$^{3+}$ (DMEM + Hb-Fe$^{3+}$), or A549 cells in DMEM (A549) were infected with TIGR4 and incubated for 2, 4, or 6 h. H$_2$O$_2$ concentration is per mL of media. (D-E) THY or THY containing Hb-Fe$^{3+}$ (Hb-Fe$^{3+}$) was infected with TIGR4 and incubated for 2, 4, or 6 h. As a control for D, b-Fe$^{3+}$ was left uninfected but incubated under the same conditions. Supernatants from C and E were collected, and H$_2$O$_2$ was quantified using Amplex Red. Supernatants from D were also collected to observe spectra between 350 and 500 nm via a spectrophotometer (BMG LabTech Omega). Error bars represent the standard errors of the means calculated using data from at least two independent experiments. Statistical significance was determined using a one-way analysis of variance with Tukey's post hoc test for multiple comparisons. *, $P < 0.05$; **, $P < 0.01$; ***, $P < 0.001$; ****, $P < 0.0001$.

correlation between the rise of the 305 nm-absorbing species and a decrease in the Soret peak, indicating that the oxidation of Hb-Fe$^{3+}$ by Spn-H$_2$O$_2$, catalyzes heme degradation and the formation of a heme degradation product.

To further investigate whether this 305 nm species could be a Spn-secreted product when incubated with Hb-Fe$^{3+}$, we evaluated the formation of the ~305 nm-absorbing species in THY-Hb-Fe$^{3+}$ cultures of hydrogen peroxide-deficient mutants TIGR4Δ*spxB*Δ*lctO*, TIGR4Δ*spxB*, or TIGR4Δ*lctO* or complemented strains. As shown in Fig. 4B, the 305 nm-absorbing species was present in lower concentrations in cultures of TIGR4Δ*spxB*Δ*lctO* and TIGR4Δ*spxB*, but its formation was observed in THY-Hb-Fe$^{3+}$ cultures of TIGR4Δ*lctO* and complemented strains TIGR4Δ*lctO*Ω*lctO* and TIGR4Δ*spxB*Δ*lctO*Ω*spxB*, indicating that the 305 nm species does not arise from a molecule secreted by Spn. Moreover, these results confirm that Spn-H$_2$O$_2$ catalyzed the formation of this species (Fig. 4B). The formation of the 305 nm-absorbing species was

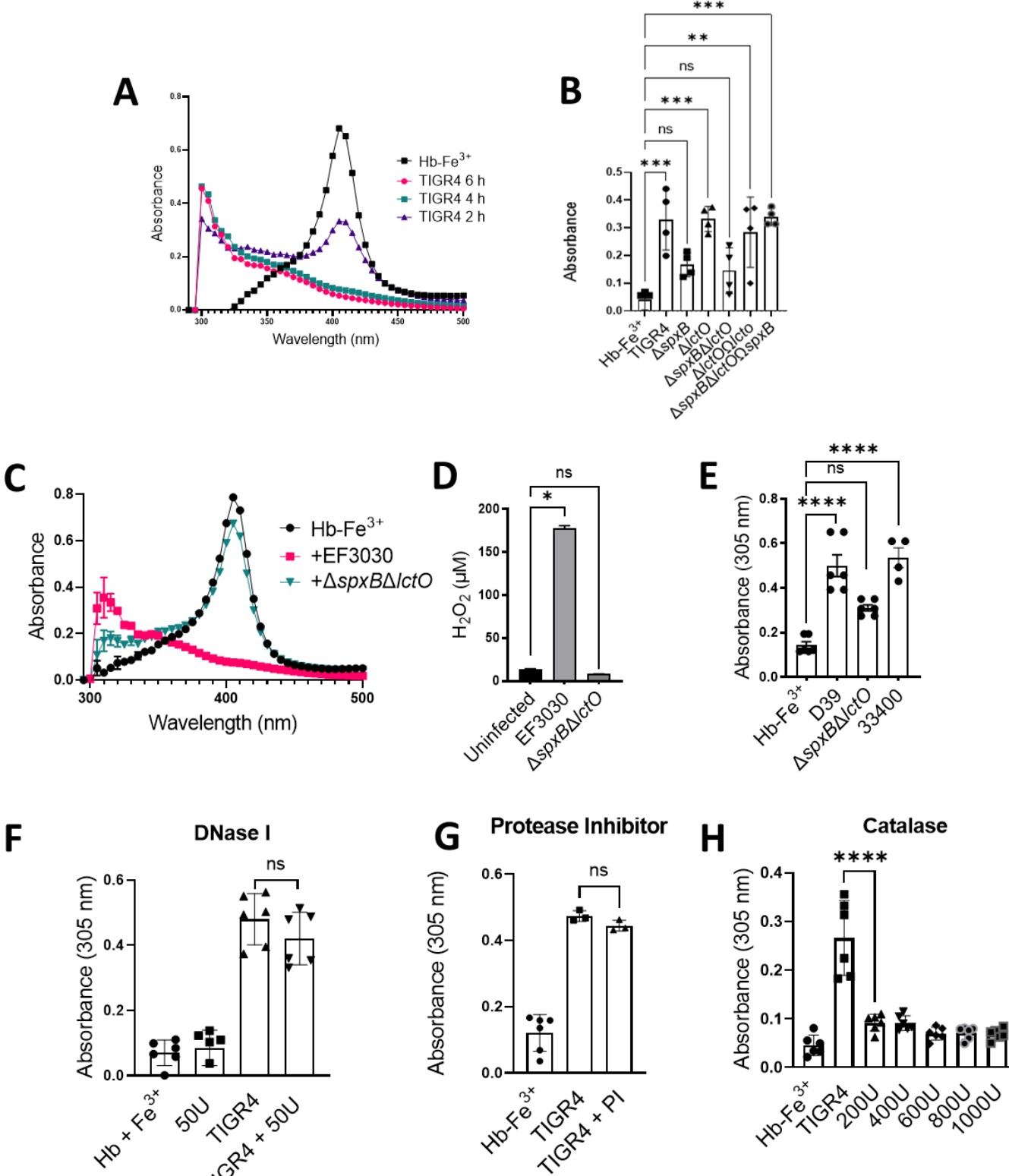

**FIG 4** A $H_2O_2$-dependent molecule is formed in cultures of Spn with Hb-$Fe^{3+}$. (A, C, and E) THY containing Hb-$Fe^{3+}$ (Hb-$Fe^{3+}$) was infected with TIGR4 (A), EF3030 (C), D39 (E), and ATCC 33400 (E) and incubated for 2, 4, or 6 h. As a control, Hb-$Fe^{3+}$ was left uninfected but incubated under the same conditions. Supernatants were collected, and the spectra between 250 and 500 nm were obtained using a spectrophotometer (BMG LabTech Omega). (B) TIGR4, its mutant derivative, or complemented strains were cultured under similar conditions as above for 4 h. Uninfected THY-Hb-$Fe^{3+}$ (Hb-$Fe^{3+}$) served as a control. The supernatants were

**FIG 4** (Continued)

purified and analyzed as above. The absorbance at 305 nm obtained for each assessed strain was used to construct the graph. (D) Supernatants from C were also collected for quantification of $H_2O_2$ using the Amplex Red Hydrogen Peroxide Assay. (F-H) THY-Hb-$Fe^{3+}$ (Hb-$Fe^{3+}$) was inoculated with TIGR4 and treated with (F) DNase I (50 U), (G) a cocktail of serine and cysteine protease inhibitor (PI), or (H) catalase (200, 400, 600, 800, or 1,000). As a control, Hb-$Fe^{3+}$ was left uninfected (Hb-$Fe^{3+}$) in F-H or treated with DNase I (50 U) in F. The absorbance at 305 nm obtained for each assessed treatment was used to construct the graph. Error bars represent the standard errors of the means calculated using data from at least two independent experiments. The level of significance was determined using a *t*-test. *, $P < 0.05$; **, $P < 0.01$; ***, $P < 0.001$; ****, $P < 0.0001$; ns, non-significant.

not inhibited by incubating with either a cocktail of protease inhibitors or with DNase I, but it was inhibited by treating with catalase (Fig. 4F through H). Catalase inhibited Hb-$Fe^{3+}$ oxidation, heme degradation, and the formation of the 305 nm shoulder, suggesting that this novel hemoglobin-derived species is formed through oxidative reactions catalyzed by Spn-$H_2O_2$ (Fig. 4H).

To test whether the formation of the 305 nm species could be inhibited with antioxidants, TIGR4 cultures in THY-Hb-$Fe^{3+}$ were treated with the antioxidants thiourea or hydroxyurea. Both thiourea and hydroxyurea inhibited the formation of the 305 nm species significantly in a dose-dependent manner (Fig. 5A and B). In THY-Hb-$Fe^{3+}$ cultures inoculated with TIGR4 and treated with either thiourea (40 or 60 mM) or hydroxyurea (3 or 8 mM), heme was degraded to the same extent as that of untreated TIGR4 cultures (Fig. 5A and B). Treating TIGR4 cultures with either of these concentrations of thiourea or hydroxyurea did not affect the growth of TIGR4 (Fig. 5C and D). However, both thiourea and hydroxyurea significantly inhibited the formation of the 305 nm species, suggesting

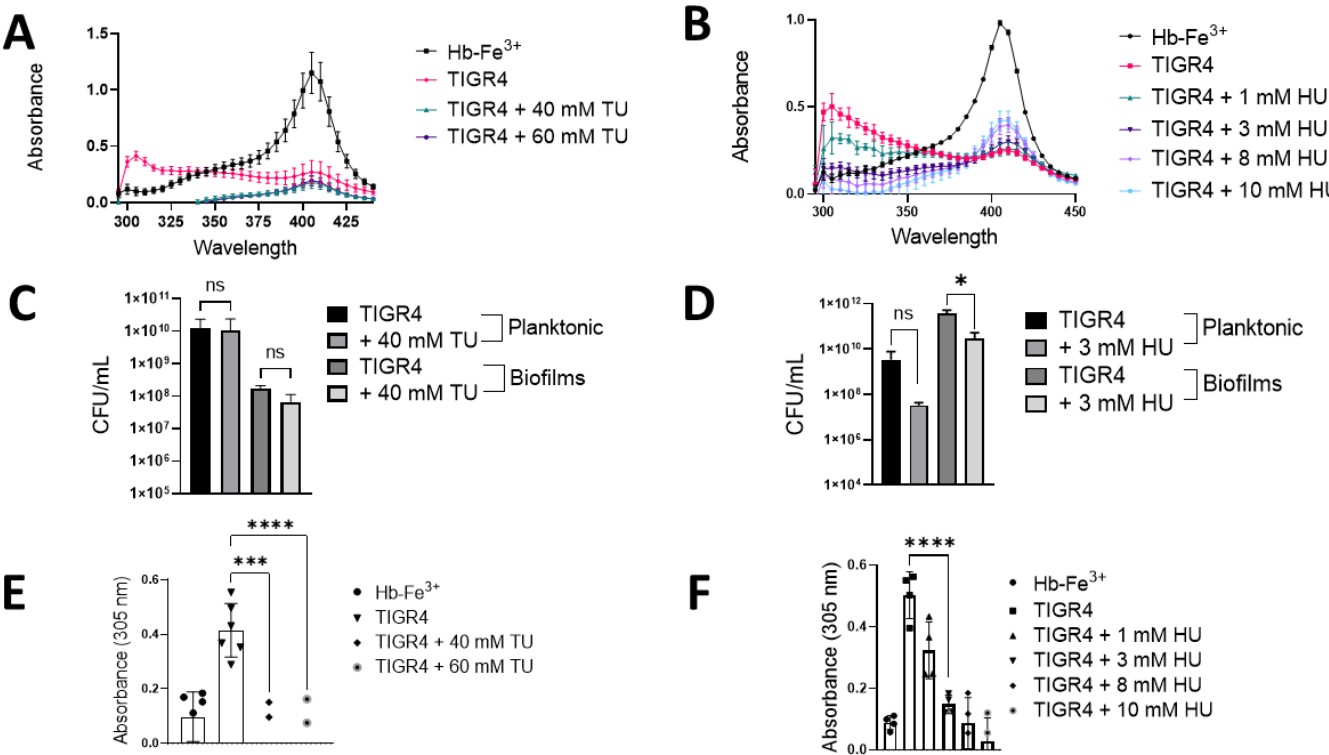

**FIG 5** A potential Hb-$Fe^{3+}$-derived molecule is catalyzed by incubation with Spn-$H_2O_2$. (A-F) THY containing Hb-$Fe^{3+}$ (THY-Hb-$Fe^{3+}$) was infected with TIGR4 and incubated for 4 h. As a control, THY-Hb-$Fe^{3+}$ was left uninfected and incubated under the same conditions as Hb-$Fe^{3+}$. TIGR4 cultures were treated with (A, C, E) thiourea or (B, D, F) hydroxyurea and incubated under the same culture conditions. (A and B) Supernatants were collected, and the spectra between 280 and 430 nm were obtained using a spectrophotometer (BMG LabTech Omega). (C and D) Planktonic bacteria or biofilms formed on the bottom of the wells were collected and plated to obtain CFU/mL density (E and F). The absorbance at 305 nm obtained for each assessed culture condition was used to construct the graph. Error bars represent the standard errors of the means calculated using data from at least two independent experiments. The level of significance was determined using a *t*-test. *, $P < 0.05$; ***, $P < 0.001$; ****, $P < 0.0001$; ns, non-significant.

that the presence of this molecule(s) is associated with the formation of a radical (Fig. 5E and F).

## The Hb-Fe$^{3+}$-derived molecule is formed in cultures of other hydrogen peroxide-producing *S. pneumoniae* strains

To investigate if the molecule found at ~305 nm is produced by other pneumococcal strains, cultures of THY-Hb-Fe$^{3+}$ were inoculated with other reference pneumococcal strains, and the infected cultures were incubated for 4 h. The ~305 nm-absorbing species was observed in cultures of EF3030, D39, and Spn reference strain ATCC33400 but not in uninfected cultures of THY-Hb-Fe$^{3+}$ (Fig. 4C and E). To further confirm that H$_2$O$_2$ caused the formation of the ~305 nm-absorbing species in cultures of strain EF3030, or strain D39, we utilized hydrogen peroxide-defective mutant EF3030Δ*spxB*Δ*lctO* or D39Δ*spxB*Δ*lctO*. We have demonstrated that EF3030Δ*spxB*Δ*lctO* does not produce H$_2$O$_2$, whereas EF3030 produced 174 µM of H$_2$O$_2$ in THY cultures incubated for 4 h (Fig. 4D). The detection of H$_2$O$_2$ activity in the supernatants of EF3030Δ*spxB*Δ*lctO* was similar to that detected in uninfected THY cultures (Fig. 4D). THY-Hb-Fe$^{3+}$ cultures were then infected with EF3030Δ*spxB*Δ*lctO* (Fig. 4C) or with D39Δ*spxB*Δ*lctO* (Fig. 4E), and the cultures were incubated for 4 h. Experiments demonstrated that the production of ~305 nm-absorbing molecules requires the secretion of hydrogen peroxide into the supernatant (Fig. 4C and E).

## Evidence for the formation of a tyrosyl radical as a byproduct of Hb-Fe$^{3+}$ oxidation by *S. pneumoniae* H$_2$O$_2$

To investigate whether Spn-H$_2$O$_2$ triggers the formation of radical species through its reaction with Hb-Fe$^{3+}$, THY was inoculated with TIGR4, EF3030, their H$_2$O$_2$ mutant derivatives (Δ*spxB*Δ*lctO*), or a complemented strain (Ω*spxB*Ω*lctO*). Hb-Fe$^{3+}$ and DEPMPO [5-(diethoxyphosphoryl)-5-methyl-1-pyrroline-N-oxide, 98%] were subsequently introduced into the supernatants, and radical generation was monitored by using continuous wave-EPR. Identical EPR spectra were observed from TIGR4, TIGR4Ω*spxB*Ω*lctO*, and EF3030 supernatants in the presence of Hb-Fe$^{3+}$, as shown in Fig. 6A. Furthermore, the addition of catalase (Fig. 6A, +Cat) eliminated the EPR signal. These findings show that the formation of radical species and spin trap adducts requires the presence of both H$_2$O$_2$ and Hb-Fe$^{3+}$. The minor signal in the THY+Hb-Fe$^{3+}$ sample arises from Hb-Fe$^{3+}$ autoxidation (ctrl red, Fig. 6A) (43).

Simulations of the EPR spectra (Fig. 6B) led to deconvolution into three components, which had similar isotropic hyperfine coupling constants from interactions of the trapped unpaired electron spin with the phosphorus, nitrogen, and β-hydrogen nuclei of DEPMPO, with mean values of $a_P = 45.6 \pm 0.7$, $a_N = 14.4 \pm 0.1$, and $a_H = 21.3 \pm 0.5$ Gauss, respectively. The 12-line EPR lineshape and isotropic hyperfine coupling constants are consistent with spin adducts formed with DEPMPO and the tyrosyl radical, possibly through the reaction at the 3- or 5-carbon position on the side chain phenolic ring (44–47). Two motional components contribute to the spectra. The predominant, relatively slow-motion component, accounting for 66% of the population, displays a more pronounced line-broadening, with a linewidth of 0.58 mT. The lineshape of this component is consistent with the trapping of a tyrosyl side chain radical on Hb-Fe$^{3+}$, which is an intermediate in the reaction of H$_2$O$_2$ with Hb-Fe$^{3+}$ (46, 48, 49). The minor, relatively fast-motion component (total, 34%) corresponds to a tyrosyl radical at a less ordered site on the protein. Formation of the tyrosyl radical by reaction of tyrosine on Hb-Fe$^{3+}$ with the previously reported ˙OH, is highly likely to cause the cytotoxicity seen in cell culture samples (8) (Fig. 3A).

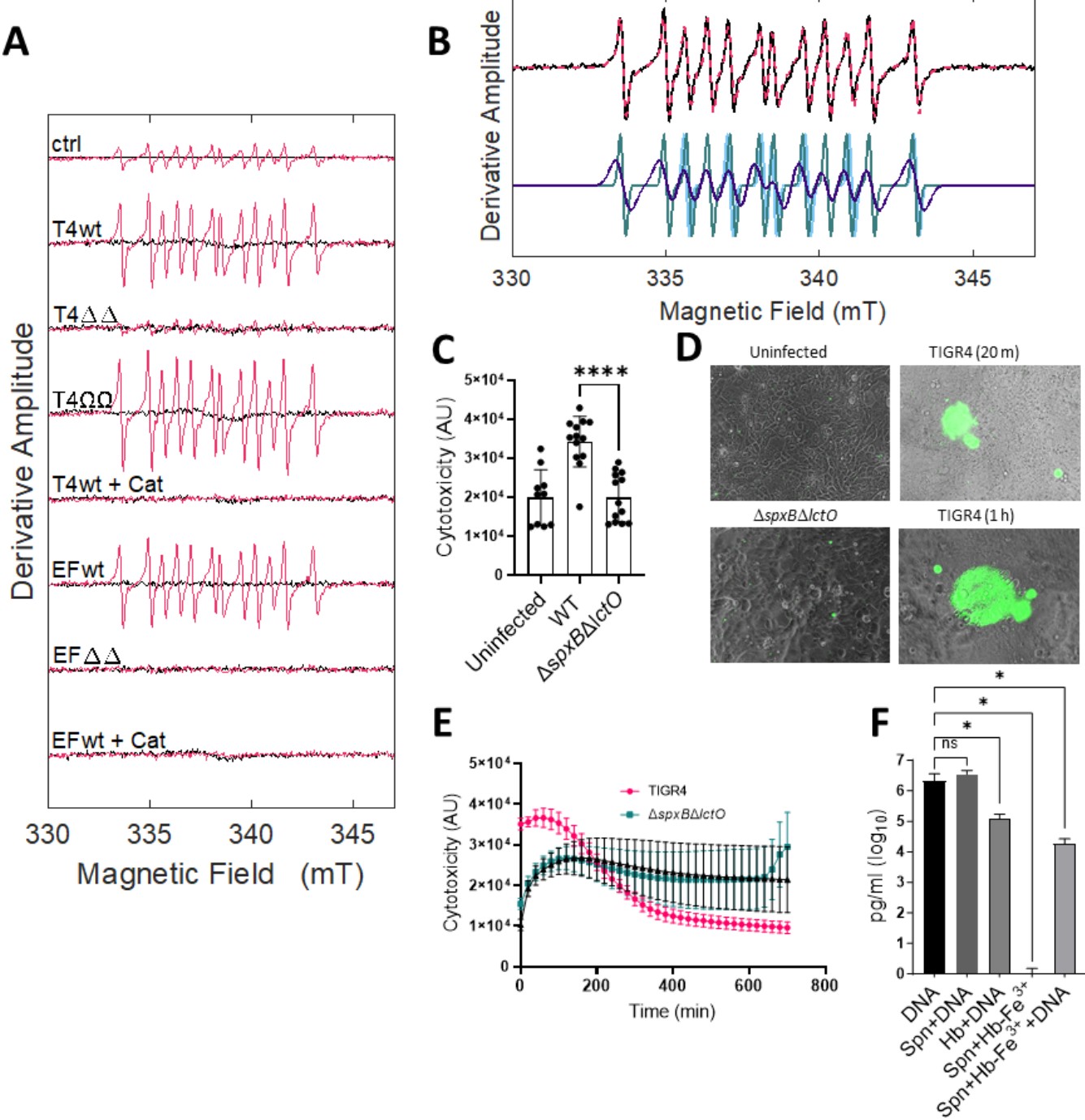

**FIG 6** Oxidation of Hb-Fe$^{3+}$ by Spn-H$_2$O$_2$ leads to the formation of radicals and cytotoxicity through DNA degradation. (A) THY was inoculated with TIGR4, EF3030, and their H$_2$O$_2$-deficient mutant derivates (ΔΔ) or H$_2$O$_2$-producing complemented strains (ΩΩ) for 4 h at 37°C, and the supernatants of these cultures were utilized for spin-trapping. Hb-Fe$^{3+}$ (10 μM) and DEPMPO (30 mM) were subsequently introduced into these supernatants to a final volume of 40 μL. Each spectrum (−Hb, black; +Hb, red) represents an average of 20 scans, minus the buffer baseline. As a control (ctrl), uninfected THY was sampled in the same manner. Spn-H$_2$O$_2$-free samples (+Cat) were obtained by pretreating the bacterial supernatant with 100 U of catalase for 1 min. (B) The 12-line EPR spectrum (T4wt, black) and overlaid simulated spectrum simulation (red) are at the top. The spectrum simulation (bottom) is the sum of three components, with the following normalized percentage contributions, isotropic hyperfine coupling constants for phosphorus, nitrogen, and hydrogen, and linewidth parameters: (i) 66% ($a_P = 45.0$, $a_N = 14.6$, $a_H = 21.7$ Gauss), 5.8 Gauss (ii); 17%, ($a_P = 46.5$, $a_N = 14.4$, $a_H = 20.7$ Gauss), 1.8 Gauss (iii); 17% ($a_P = 45.5$, $a_N = 14.3$, $a_H = 21.6$ Gauss), 1.8 Gauss. The larger linewidth of the dominant component (66%) indicates motional restriction relative to the more mobile species (34%). (C-E) Human bronchial Calu-3 cells were infected with TIGR4, its mutant derivative, and supplemented with 10 μM Hb-Fe$^{3+}$ and incubated for 24 h at 37°C in a fluorometer, obtaining (Continued on next page)

**FIG 6** (Continued)

readings every 20 min to assess cytotoxicity. (C) Cytotoxicity fluorescent readings (AU) were taken at the 20-min time interval and used to produce the graph. (D) Following 24 h of incubation, samples were imaged on an inverted microscope (Nikon Eclipse TSR), where green fluorescence indicates DNA binding and cytotoxicity of Calu-3 cells. (E) Twenty-four hours time course reading of cytotoxicity fluorescent readings (AU) of infected Calu-3 cells as described above. (F) Cultures of THY or THY supplemented with Hb-Fe$^{3+}$ (Hb-Fe$^{3+}$) were added with TIGR4 DNA (DNA) and/or inoculated with serotype 2 pneumococcal strain R6 (Spn) for 2 h, and DNA measurements (pg/mL) were obtained using qPCR. Error bars represent the standard errors of the means calculated using data from at least two independent experiments. The level of significance was determined using a $t$-test. *, $P < 0.05$; ****, $P < 0.0001$; ns, non-significant.

## Oxidized Hb-Fe$^{3+}$ causes cytotoxicity in human lung cells in part through DNA degradation

We then assessed the toxicity of Spn-H$_2$O$_2$ on polarized cultures of human bronchial Calu-3 cells by infecting them with pneumococci and Hb-Fe$^{3+}$. Cytotoxicity was evaluated using a more sensitive CellTox Green Cytotoxicity Assay (CellTox). The CellTox assay utilizes a fluorophore that binds nuclear DNA when cell membranes are permeable due to cytotoxicity, and the fluorescence can be collected in real-time. Toxicity was observed 20 min post-inoculation of bronchial lung cells with TIGR4 in the presence of hemoglobin, whereas cells infected with TIGR4Δ*spxB*Δ*lctO* yielded comparable levels of low toxicity to that of the uninfected, Hb-Fe$^{3+}$-treated negative control (Fig. 6C and D). CellTox staining was characterized by localized fluorescence, suggesting that pneumococci-forming microcolonies induce rapid membrane permeabilization. Toxicity levels of bronchial Calu-3 cells infected with TIGR4 in the presence of Hb-Fe$^{3+}$ remained significantly different, compared to the other infection conditions, during the first hour post-infection, after which the fluorescence levels declined to levels seen in the negative control (Fig. 6E).

It is well known that H$_2$O$_2$-derived radicals attack at the sugar or the base of the DNA, leading to strand breaks at a terminal fragmented sugar residue (8, 50). We hypothesize that the toxic tyrosyl radical, found above in EPR experiments, formed during the oxidation of Hb-Fe$^{3+}$ by Spn-H$_2$O$_2$ in cultures of bronchial cells infected with TIGR4 and caused DNA breaks. To test this hypothesis, DNA from TIGR4 was added to THY-Hb-Fe$^{3+}$ and infected with another H$_2$O$_2$-producing Spn strain, R6 (13, 19). The exogenous TIGR4 DNA was purified from the supernatants, and levels were compared with those of uninfected THY cultures added with TIGR4 DNA, R6-infected THY cultures added with TIGR4 DNA, THY-Hb-Fe$^{3+}$ cultures added with TIGR4 DNA, or R6-infected THY-Hb-Fe$^{3+}$ cultures lacking externally added TIGR4 DNA. A serotype-specific qPCR reaction targeting TIGR4 DNA (serotype 4) was used to differentiate the externally added DNA from traces of extracellular DNA from R6 bacteria (serotype 2), secreted by autolysis into the supernatant (34). After 2 h of incubation, a median of $2.14 \times 10^6$ pg/mL of DNA was obtained in THY added with DNA, whereas a significant reduction ($1.81 \times 10^4$ pg/mL) of the DNA level was observed in THY-Hb-Fe$^{3+}$ cultures infected with R6 (Fig. 6F). In the absence of pneumococci, Hb-Fe$^{3+}$ was able to degrade DNA, perhaps due to its autoxidation, whereas no DNA was detected in those cultures lacking TGR4 DNA, suggesting that the reaction did not amplify DNA from R6 (Fig. 6F). As well, there was no decrease in DNA detected in cultures of R6 in the absence of Hb-Fe$^{3+}$, indicating that H$_2$O$_2$ alone did not significantly degrade the externally added DNA (Fig. 6F). Collectively, these results demonstrate that the oxidation of Hb-Fe$^{3+}$ caused increased cytotoxicity to human bronchial cells, in part due to the DNA degradative capacity of Spn-induced oxidation of Hb-Fe$^{3+}$.

## *S. pneumoniae*-produced H$_2$O$_2$ oxidizes cytochrome $c$ and leads to mitochondrial respiratory dysfunction

Another important intracellular hemoprotein is Cyt$c$, which is an essential component of the respiratory chain. Cyt$c$ is also a key mediator of apoptosis, where this activity is directly related to its oxidation in the inner membrane space and subsequent release into the cytoplasm (24). Therefore, we investigated if Spn-H$_2$O$_2$ oxidizes Cyt$c$. To assess

this, THY was supplemented with 56 µM Cyt*c* THY-Cyt*c*, inoculated with TIGR4 or TIGR4Δ*spxB*Δ*lctO*, and incubated under a 5% $CO_2$ atmosphere for 6 h. The supernatants were analyzed by spectroscopy. As hypothesized, a heme Soret peak, ~415 nm, was observed in supernatants from the uninfected control, whereas in culture supernatants of TIGR4, the Soret band was absent (Fig. 7A, dark purple arrow), indicating its oxidation and suggesting heme degradation. Alpha and beta chains (Fig. 7A, light blue arrows) seen in the uninfected control were absent in TIGR4 supernatants, further indicating Cyt*c* oxidation. The ~305 nm-absorbing species was also observed in these culture supernatants (Fig. 7A, light purple arrow), suggesting that the reaction between Spn-$H_2O_2$ and Cyt*c* produces oxyferryl heme intermediates observed earlier with exogenous $H_2O_2$ (51). In contrast, in THY-Cyt*c* cultures of TIGR4Δ*spxB*Δ*lctO*, the heme moiety produced a Soret peak similar to that in the uninfected control, and Cyt*c* was not oxidized.

A qualitative in-gel heme detection assay was utilized to confirm that heme had been degraded in THY-Cyt*c* cultures of TIGR4. As illustrated in Fig. 7B and C, heme-bound to Cyt*c* was observed at ~15 kDa, and consistent with its hexacoordinate state, free heme was not observed in uninfected THY-Cyt*c* cultures. Heme-bound to Cyt*c*, however, was not detected in supernatants of TIGR4, EF3030, and D39 cultures at 6 h but remained intact in those from TIGR4Δ*spxB*Δ*lctO*, EF3030Δ*spxB*Δ*lctO,* and D39Δ*spxB*Δ*lctO* (Fig. 7B and C). Therefore, Spn-$H_2O_2$ releases heme from Cyt*c* and causes its degradation.

Under physiologic and pathophysiologic conditions, the formation of $H_2O_2$ (<50 µM) inactivates NADH-linked state 3 respiration (ADP-dependent) (52); however, this appears to be due to alterations in the steady-state levels of reducing equivalents from the Krebs cycle that are required for electron transport. Given that Spn can produce 20-fold more $H_2O_2$ (~1 mM) and this concentration oxidizes and degrades Cyt*c*, an essential component in the mitochondrial respiratory chain, mitochondrial function in the presence of Spn-$H_2O_2$ was investigated. Mitochondria were isolated from the rat heart and assessed for their functionality through observation of oxygen consumption, indicating electron

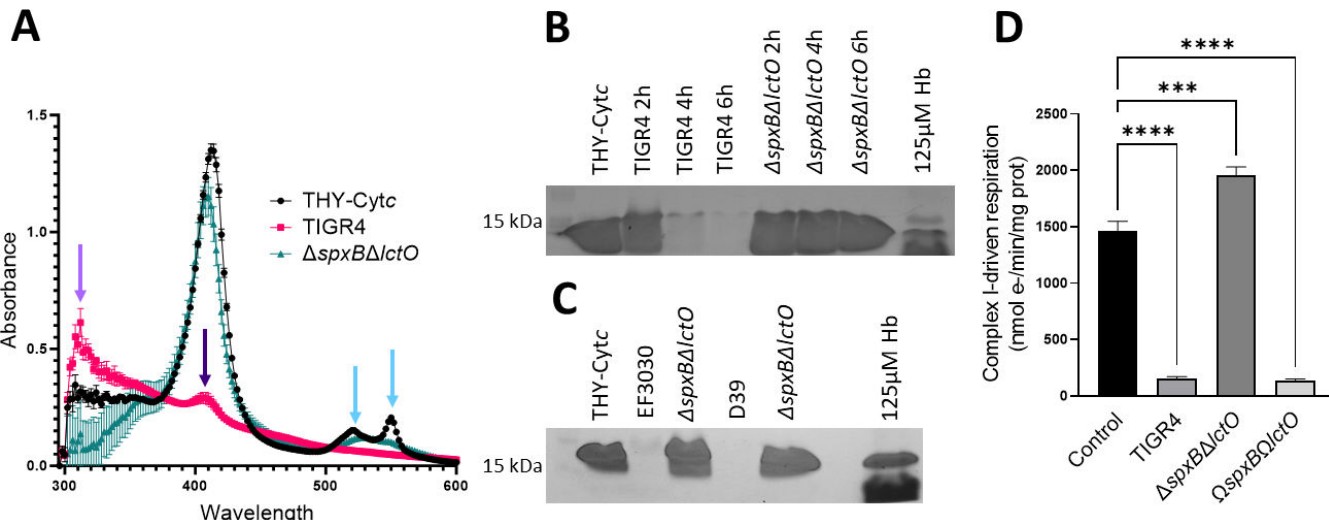

**FIG 7** Spn-$H_2O_2$ oxidizes cytochrome *c* and causes mitochondrial respiratory dysfunction. (A) THY containing 56 µM Cyt*c* was infected with TIGR4 and its mutant derivative and incubated for 4 h. As a control, Cyt*c* was left uninfected but incubated under the same conditions (THY-Cyt*c*). Supernatants were collected, and the spectra between 250 and 500 nm were obtained using a spectrophotometer, a BMG LabTech Omega. (B) Supernatants cultured in the same manner as A but with 28 µM Cyt*c* and for 2, 4, or 6 h were used for detection of heme in an in-gel heme detection assay. As a control, 125 µM Hb-$Fe^{3+}$ was used as a hemoprotein control. (C) EF3030, D39, and their mutant derivatives were cultured for 6 h as mentioned in A and stained for heme as mentioned in B. (D) Isolated intact heart mitochondria were treated with 100 µL of sterilized supernatants from either TIGR4, its mutant derivative, or a complemented strain grown for 4 h. Complex I-driven respiration was measured in an Oroboros O2K FluoRespirometer using 20 mM glutamate and 10 mM malate as substrates with the addition of 2 mM ADP. Samples were normalized to protein content as determined by the BioRad DC assay and reported as nmol e-/min/mig prot. Error bars represent the standard errors of the means calculated using data from six independent studies. The level of significance was determined using a *t*-test. ***, $P < 0.001$; ****, $P < 0.0001$.

transfer and ATP production. Using an Oroboros O2k FluoRespirometer, electron transfer was measured through complex I-driven respiration, which proceeds electron movement through complexes I, III, and IV. Isolated heart mitochondria exposed to uninfected THY did not show disruption of electron transport to oxygen through complex I-driven respiration (Fig. 7D). In contrast, supernatants containing $H_2O_2$ harvested from cultures of TIGR4 or the complemented strain $\Omega spxB\Omega lctO$ caused a significant decrease in complex I-driven respiration (Fig. 7D). Mitochondria incubated with supernatants from cultures of $H_2O_2$-deficient $\Delta spxB\Delta lctO$ pneumococci did not show an effect on the transference of electrons to oxygen through complex I-driven respiration (Fig. 7D).

## DISCUSSION

Cytotoxicity caused by *Streptococcus pneumoniae*-produced hydrogen peroxide (Spn-$H_2O_2$) has been well established, and evidence includes cytotoxic activity against alveolar cells, cardiomyocytes, and explants of human ciliated epithelium (3, 6, 53). However, a gap currently remains in identifying the specific molecular target(s) of Spn-$H_2O_2$ that causes toxicity during pneumonia infection. In this study, we have recapitulated critical steps of the pathophysiology of pneumococcal pneumonia, including infection, colonization, and invasion of lung bronchial and alveolar cells, and find that Spn-$H_2O_2$ is produced inside lung cells by invading pneumococci to increase ROS and to oxidize hemoproteins. The oxidation of hemoproteins led to the collapse of the cell's actin cytoskeleton, loss of the microtubule's cytoskeleton, degradation of bronchial cellular DNA, and disruption of mitochondrial function, ultimately causing cell death. Apoptosis and, more recently, pyroptosis have been implicated in the mechanism of cell death induced by Spn-$H_2O_2$ (4, 16). In human bronchial 16HBE cells, Spn-$H_2O_2$ caused the activation of caspase 3/7, a hallmark of apoptosis, but it also caused the activation of caspase 1, which is a characteristic of pyroptosis (16). The pneumolysin Ply has also been implicated in apoptosis (3, 53); however, Surabhi and colleagues demonstrated that in the early stages of infection (i.e., within 6 h post-inoculation), Spn-$H_2O_2$ was sufficient to trigger apoptosis and pyroptosis (16). These two cell death pathways may be activated by Spn-$H_2O_2$ simultaneously, suggesting the disruption of a number of different intracellular targets.

Correlating with previous studies indicating that $H_2O_2$ produced through the catalytic reaction of SpxB causes cytotoxicity of human bronchial 16HBE cells (16), our double *spxB* and *lctO* knockout and single *spxB* knockout were significantly attenuated for causing cytotoxicity of human lung alveolar A549 cells and human bronchial Calu-3 cells within 6 h post-inoculation. We further demonstrated that the cytotoxicity was mainly caused by the formation of a toxic radical, since there was no detectable Spn-$H_2O_2$ in lung cell cultures infected with $H_2O_2$-producing pneumococci. In support of this, we have demonstrated, by using EPR spectroscopy, the presence of a relatively immobile radical, assigned to a protein tyrosyl-DEPMPO adduct radical, indicating that the tyrosyl radical is produced through the oxidation of Hb-$Fe^{3+}$ by Spn-$H_2O_2$.

How could such a tyrosyl radical be formed inside lung cells? It is known that a tyrosyl radical can be formed upon exposure of oxygen to metmyoglobin (46, 48), Cyt*c* (54, 55), and now methemoglobin (Hb-$Fe^{3+}$). In myoglobin, the globin-centered free radical was detected at Tyr-103, whereas a tyrosyl radical in Cyt*c* was identified in three solvent-exposed tyrosine residues (54). There are three tyrosine residues in the α-chain of hemoglobin (Tyr-24, Tyr-42, and Tyr-140) and three (Tyr-35, Tyr-130, and Tyr-145) in the β-chain, although those in the latter are solvent-exposed and therefore more prone to radical formation. There are several lines of evidence that led us to formulate our hypothesis. First, we demonstrated peroxidase activity when cultures of human cells were infected with Spn, and this activity correlated with cytotoxicity. Subsequently, EPR studies identified the presence of a toxic radical during pseudoperoxidase (i.e., Hb-$Fe^{3+}$) reactions. Additionally, we demonstrated that, at the time cytotoxicity occurred, there was a concurrent increase in the detection of ROS species inside lung cells. This correlates with recent evidence from our laboratories, which revealed that Spn-$H_2O_2$

oxidizes Hb and Cyt*c* (this study), releasing iron (13, 19). As a result, the findings from our experiments support a scenario where the oxidation of intracellular hemoproteins by the production of abundant $H_2O_2$ by intracellular pneumococci leads to the formation of a toxic tyrosyl radical.

During the pathophysiology of pneumococcal pneumonia, invasion of the bronchi and terminal bronchiole leads pneumococci to reach the respiratory bronchiole, alveolar ducts, and alveoli, where pneumococci have access to the alveolar-capillary network, a large surface area containing ~300 billion blood capillaries where the exchange of $CO_2$ for $O_2$ occurs. Moreover, pneumococcal pneumonia is characterized by pulmonary hemorrhage, inflammatory congestion, and hepatization, leading to lung parenchymal injury (56). Pneumococci are therefore exposed to Hb in the lung parenchyma, and we have demonstrated that cultures of pneumococci oxidize Hb, and EPR experiments with Hb-$Fe^{3+}$ incubated with exogenous $H_2O_2$ identified a tyrosine carbon radical oxidation product. When $H_2O_2$-producing pneumococci infected human lung alveolar cells growing in the presence of Hb-$Fe^{3+}$, cytotoxicity occurred as early as 20 min post-inoculation. Correlating with cytotoxicity, degradation of bronchial cellular DNA was also observed. This cytotoxicity and DNA degradation were unable to be directly attributed to Spn-$H_2O_2$, as the oxidant could not be directly detected in culture supernatants. Thus, this suggests potential pseudo-peroxidase and endonuclease activity in Hb-$Fe^{3+}$-containing supernatants, which leads to the formation of toxic radicals.

Correlating with pseudoperoxidase activity, Hb-$Fe^{3+}$ was oxidized in culture supernatants, as shown by the loss of UV-visible absorbance at ~405 nm, with the formation of intermediate, or degradation products, that were observed by spectroscopy as a rise in absorbance at shorter wavelengths, with a shoulder at 305 nm. Western blot analysis of the supernatants with an anti-Hb antibody could not detect these degradation products (not shown). However, the formation of such intermediate products was only inhibited by the radical scavengers hydroxyurea and thiourea, indicating that the intermediate product(s) are the result of non-specific proteolysis. Given that treating Hb-$Fe^{3+}$-containing cultures with catalase also inhibited the formation of the ~305 nm species, we propose that the radical was derived from Hb-$Fe^{3+}$. This is consistent with the formation of a tyrosyl radical as an intermediate in the reaction of $H_2O_2$ with Hb-$Fe^{3+}$ (46, 48). The quenching of the potential radical by hydroxyurea may lead to an alternative, or improved, therapeutic approach for pneumococcal pneumonia, as this compound is already in use for patients with sickle cell disease and β-thalassemia (57–60).

To further attest that oxidation of other hemoproteins occurs, the degradation of heme by Spn-$H_2O_2$ also occurred with a major mitochondrial hemoprotein, Cyt*c*. Spn-$H_2O_2$ is known to cause mitochondrial damage, but the direct target of Spn-$H_2O_2$ on the mitochondria is unknown (15). As mitochondria are a major target of oxidative compounds and contain an abundant amount of both Cyt*c* and heme, the impact of Spn-$H_2O_2$ on mitochondrial function was observed. Utilizing Spn supernatants containing up to ~700 μM $H_2O_2$, we observed the loss of complex I-driven mitochondrial function. Respiration driven by complex I involves electron transport through complexes I, III, and IV. These complexes are characterized by their iron-sulfur (Fe-S) or heme centers and have the potential to be oxidized by Spn-$H_2O_2$, resulting in the loss of mitochondrial respiration. Understanding how Spn-$H_2O_2$ is affecting mitochondrial function will provide additional insight into disease progression.

In summary, this study demonstrates that Spn-$H_2O_2$ plays a major role in the cytotoxicity of human lung cells through the oxidation of hemoproteins, key proteins in all biological processes in the cell. We described that the pseudoperoxidase activity of Hb-$Fe^{3+}$, and perhaps that of Cyt*c*, is associated with the formation of the tyrosyl radical that causes cytotoxicity. Cell death induced by the oxidation of hemoproteins by Spn-$H_2O_2$ leads to a mechanism with signs of apoptosis and pyroptosis that also results in the collapse of the actin and microtubule cytoskeletons, DNA degradation, and disruption of mitochondrial function. The specific mechanisms leading to cytoskeletal loss and mitochondrial dysfunction, as well as how pneumococci activate the

production of Spn-$H_2O_2$ as part of their intracellular lifestyle, are under investigation in our laboratory.

## ACKNOWLEDGMENTS

This study was supported in part by grants from the National Institutes of Health (NIH; 5R21AI144571-03) to J.E.V. and (NIH; 9R01GM142113 and RR17767) to K.W. B.A. was supported by a Fulbright scholarship awarded by the US Department of State. Studies of confocal microscopy and mitochondrial respiration (KE) were supported by a grant from the National Institute of General Medical Sciences (NIGMS) of the National Institutes of Health under award number P20GM121334. J.E.V. is also supported by a grant from NIGMS through the Molecular Center of Health and Disease (1P20GM144041-01A1 7651).

The content is solely the responsibility of the authors and does not necessarily represent the official view of the NIH, the US Department of State, or the Department of Cell and Molecular Biology of the University of Mississippi Medical Center (UMMC).

The authors thanks Dr. David Brown from UMMC for his assistance with confocal microscopy.

## AUTHOR AFFILIATIONS

[1]Department of Cell and Molecular Biology, School of Medicine, University of Mississippi Medical Center, Jackson, Mississippi, USA

[2]Center for Immunology and Microbial Research, School of Medicine, University of Mississippi Medical Center, Jackson, Mississippi, USA

[3]Department of Otolaryngology-Head and Neck Surgery, The Ohio State School of Medicine, The Ohio State Wexner Medical Center, Columbus, Ohio, USA

[4]Mississippi INBRE Research Scholar, University of Southern Mississippi, Jackson, Mississippi, USA

[5]Department of Physics, Emory University, Atlanta, Georgia, USA

## AUTHOR ORCIDs

Anna Scasny http://orcid.org/0000-0003-3567-1157
Jorge E. Vidal http://orcid.org/0000-0003-0573-5658

## FUNDING

| Funder | Grant(s) | Author(s) |
|---|---|---|
| HHS \| NIH \| National Institute of Allergy and Infectious Diseases (NIAID) | 5R21AI144571-03 | Jorge E. Vidal |
| HHS \| NIH \| National Institute of General Medical Sciences (NIGMS) | 1P20GM144041-01A1 7651 | Jorge E. Vidal |
| HHS \| NIH \| National Institute of General Medical Sciences (NIGMS) | P20GM121334 | Kristin Edwards |
| | | Jorge E. Vidal |
| HHS \| National Institutes of Health (NIH) | 9R01GM142113 | Kurt Warncke |
| HHS \| National Institutes of Health (NIH) | RR17767 | Kurt Warncke |

## AUTHOR CONTRIBUTIONS

Anna Scasny, Conceptualization, Formal analysis, Investigation, Methodology, Writing – original draft, Writing – review and editing | Babek Alibayov, Formal analysis, Investigation, Supervision, Writing – review and editing | Faidad Khan, Investigation | Shambavi J. Rao, Investigation | Landon Murin, Investigation | Ana G. Jop Vidal, Investigation, Methodology, Project administration | Perriann Smith, Investigation | Wei Li, Investigation | Kristin Edwards, Formal analysis, Investigation, Methodology, Writing – review and editing | Kurt Warncke, Formal analysis, Investigation, Methodology, Writing – review

and editing | Jorge E. Vidal, Conceptualization, Data curation, Formal analysis, Funding acquisition, Investigation, Writing – original draft, Writing – review and editing

## ETHICS APPROVAL

All procedures were conducted in accordance with the National Institutes of Health's Guide for the Care and Use of Laboratory Animals and approved by the UMMC Institutional Animal Care and Use Committee (Protocol #1582).

## ADDITIONAL FILES

The following material is available online.

Open Peer Review

**PEER REVIEW HISTORY (review-history.pdf).** An accounting of the reviewer comments and feedback.

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
