## [Reviewer comments · Microbiology Spectrum]

Microbiology Spectrum

Oxidation of hemoproteins by *Streptococcus pneumoniae* collapses the cell cytoskeleton and disrupts mitochondrial respiration leading to cytotoxicity of human lung cells

Anna Scasny, Babek Alibayov, Faidad Khan, Shambavi Rao, Landon Murin, Ana Vidal, Perriann Smith, Wei Li, Kristin Edwards, Kurt Warncke, and Jorge Vidal

Corresponding Author(s): Jorge Vidal, The University of Mississippi Medical Center

Review Timeline:

Submission Date:	July 24, 2023
Editorial Decision:	September 1, 2023
Revision Received:	October 11, 2023
Accepted:	November 6, 2023

Editor: Fernando Navarro-Garcia

Reviewer(s): Disclosure of reviewer identity is with reference to reviewer comments included in decision letter(s). The following individuals involved in review of your submission have agreed to reveal their identity: Srinivasan Velusamy (Reviewer #1)

Transaction Report:

DOI: <https://doi.org/10.1128/spectrum.02912-23>

September 1, 2023

Dr. Jorge E. Vidal
The University of Mississippi Medical Center
Cell and Molecular Biology
2500 North State Street
Jackson, MS 39216

Re: Spectrum02912-23 (Oxidation of hemoproteins by *Streptococcus pneumoniae* collapses the cell cytoskeleton and disrupts mitochondrial respiration leading to cytotoxicity of human lung cells)

Dear Dr. Jorge E. Vidal:

Link Not Available

Sincerely,

Fernando Navarro-Garcia

Journals Department
Reviewer comments:

Reviewer #1 (Comments for the Author):

Oxidation of hemoproteins by *Streptococcus pneumoniae* collapses the cell cytoskeleton and disrupts mitochondrial respiration leading to cytotoxicity of human lung cells
Scasny et al.

The authors have studied mitochondrial disruption and cell cytoskeleton collapse caused by hydrogen peroxide released by Spn infecting lung cells. The authors have used appropriate methodology to show the role of H₂O₂ in cytotoxicity of human lung cells and presented with appropriate figures and tables. There are no major comments on the findings. However, there are some minor comments.

1. The length of the manuscript is very long and suggested to shorten the introduction and materials and methods sections.
2. The authors have used already published methods to validate their hypothesis and it is suggested to use the references instead of providing detailed methodology.
3. Figure numbers are missing in each page in the bottom of the manuscript. It was difficult to correlate/interpret the figures with the text.

Reviewer #2 (Comments for the Author):

Comments for the authors:

The authors have interestingly revealed possible links between H₂O₂ produced by *Streptococcus pneumoniae* and the associated cytotoxicity of it. Pneumococcal WT lab strains and mutants producing little H₂O₂ were used to compare differences in cytotoxicity, and cytotoxicity was linked to oxidation of Hb-Fe³⁺. The topic is relevant, as it could contribute to understanding the progression of invasive disease, and resulting in new therapeutics options. There are some major concerns which will be addressed below.

Major comment

There is no data shown or mentioned regarding the differences in growth between the strains, nor is the amount of bacteria present at most measurements known to the reader (and the authors?). Since the authors mention that the both SpxB and LctO are part of a pathway that is responsible for the bulk of the ATP produced. The loss of these genes is likely to affect growth to a considerable degree, resulting in different bacterial loads between the WT and mutant conditions. Whether the results observed are majorly affected by these differences is presently impossible to determine, while it could be a very important factor. Different growth dynamics, might also result in a different ratio of live:dead bacteria at the time of measurement, which could release all sorts of molecules into the growth medium which could also affect the results. CFU/DNA copy numbers, or the addition of growth curves for the used strains (as supplementary material) would greatly improve the interpretation of the shown data. Intracellular growth might be quite different and difficult to measure, but in the data as presented, it is unclear what the dynamics of bacterial growth are. The observed difference in cytotoxicity could be simply because of differences in the bacterial load.

Minor comments

Many experiments are performed with 10 μM Hb-Fe³⁺, which is shown to oxidize and result in cytotoxicity. However, most Hb in vivo would be Hb-Fe²⁺ to start with, which would first need to be oxidized to Hb-Fe³⁺ (which has been observed and reported previously). Hb-Fe³⁺ could then be oxidized to Hb-Fe⁴⁺. In reference 19, most Hb-Fe²⁺ is stated to be oxidized into Hb-Fe³⁺ in experimental conditions. It is unclear to me how much of the Hb-Fe²⁺ would be expected to be oxidized twice. So whether this oxidation of Hb-Fe³⁺ that is shown, would occur in physiological conditions. On what do the authors base the idea that Hb-Fe³⁺ oxidation could be expected in invasive disease and would it not be more logical to add Hb-Fe²⁺, which, if it also occurs in in vivo conditions, could be oxidized twice and also form the radical as observed?

Line 78: The authors mention Spx and LctO as the only enzymes that produce H₂O₂, with percentages shown in line 82-83.

There is at least one more enzyme, and possibly more, that also produce H₂O₂: GlpO, involved in the glycerol metabolic process. GlpO catalyzes the following reaction: O₂ + sn-glycerol 3-phosphate = dihydroxyacetone phosphate + H₂O₂. As the mutants show, this likely does not contribute much to the overall H₂O₂ production by Spn, so it has little consequence for the content for the paper, but the statement that these two enzymes are solely responsible for H₂O₂ production is incorrect.

Line 158: Choose either using or with

Line 182: Washed once and then collected? Or just a single addition of PBS? Probably the latter, but the word washed confused me.

Line 191: Why was this concentration of Hb used? Does this have any physiological basis? It is used throughout the paper. Although it is possibly difficult to determine the level of Hb the pneumococcus is exposed to, it is important to understand the relevance of this amount of Hb in the in vivo situation.

Line 224-226: where the nuclei counted manually or using software algorithms?

Line 239: Why was the 100 μM H₂O₂ concentration used? Especially since you mention in line 379-380 that the TIGR4 strain used produces 700 μM within 6h of incubation.

Line 383-384: What was the conclusion?

Line 418: 8h for TIGR4 but 6h for EF3030

Line 434-435: H₂O₂ itself is already a ROS. It increases other intracellular ROS?

Line 439-440: Does it not just visualize an increase of intracellular H₂O₂? That the mutants do not produce? Have you tested the same experiment with just extracellular H₂O₂ in similar concentration to see whether the observed ROS is not just the increased H₂O₂? H₂O₂ alone did not affect tubulin, suggesting another actor, but a control for H₂O₂ would have been more correct.

Line 442-444: Have you quantified this observation?

Line 446-447: If H₂O₂ is the cause of cytotoxicity, why did the 100 μM H₂O₂ not affect cytoskeleton, why were no other concentrations tested to see if this is the result of just H₂O₂ or if other factors are involved? Should the oxidation of cell molecules also not be induced by the extracellularly added 100 μM H₂O₂.

Line 459-460: This is worth showing

Line 481: Or that the amount of H₂O₂ produced (+/- 200 μM, fig. 3E) was sufficient for the cytotoxicity seen, especially since the

cytotoxicity dropped from ~25% in figure 1B in 6 hours to 7% in 14 hours (3A)? Perhaps this experiment is only showing that Hb-Fe³⁺ can reduce H₂O₂.

Line 485: It is also possible the cells reduced the produced H₂O₂, although increased catalase would have been expected in the TIGR4 16h condition in fig 1G.

Line 516: You show that growth is significantly affected in biofilm in 5D, and although not significant, over a 100-fold lower CFU in the HU condition is quite a lot less.

Line 535: Typo of *S. pneumoniae*

Line 543-544: Would you not expect more of a signal in the mutants than as well? There is some in the TIGR4 double mutant, but nothing in the EF30303 mutant.

Line 586-587: It does not necessarily show that the DNA damage that is observed is part of the cause of the cytotoxicity. It would also be interesting to add a bacteria but no Hb-Fe³⁺, as well as a double mutant, as a control to see whether the damage is caused by oxidized components, H₂O₂ or the bacteria itself. Was the H₂O₂ concentration measured in the growth of R6 with Hb-Fe³⁺? Otherwise the DNA damage could still be a result of H₂O₂.

Line 593: Again, what is this concentration based off?

Figure 1:

Showing the results of statistical comparison between two conditions in a graph seems to be chosen somewhat randomly, as in 1I for example.

1C: Were CFU counts/ODs measured to determine if the bacterial load was similar? Otherwise the effect could be due to the bacterial growth being affected by the catalase. If it is cytotoxic for A549 cells in those concentrations, perhaps it is the same for the pneumococci.

1D/1H: Why were two different timepoints chosen for the different strains? Additionally, there seem to be no bacteria in the TIGR4 DAPI stain?

1G: Would an effect of H₂O₂ on tubulin not be expected if that is what is the main difference between WT and the double mutant? The expected concentration is different though than the ~700μM that was mentioned previously. The authors do not make note of this observation, or try to explain what is seen.

Figure 2: Legend repeats part of Fig. 1 legend.

Staff Comments:

Preparing Revision Guidelines

Please return the manuscript within 60 days; if you cannot complete the modification within this time period, please contact me. If you do not wish to modify the manuscript and prefer to submit it to another journal, please notify me of your decision immediately so that the manuscript may be formally withdrawn from consideration by Microbiology Spectrum.

Comments for the editor:

Thank you for the opportunity to review "Oxidation of hemoproteins by *Streptococcus pneumoniae* collapses the cell cytoskeleton and disrupts mitochondrial respiration leading to cytotoxicity of human lung cells" for the journal

I am not experienced in EPR and therefore I am limited in the ability to comment on the methods and materials used for this part of the paper.

Comments for the authors:

The authors have interestingly revealed possible links between H_2O_2 produced by *Streptococcus pneumoniae* and the associated cytotoxicity of it. Pneumococcal WT lab strains and mutants producing little H_2O_2 were used to compare differences in cytotoxicity, and cytotoxicity was linked to oxidation of Hb-Fe³⁺. The topic is relevant, as it could contribute to understanding the progression of invasive disease, and resulting in new therapeutics options. There are some major concerns which will be addressed below.

Major comment

There is no data shown or mentioned regarding the differences in growth between the strains, nor is the amount of bacteria present at most measurements known to the reader (and the authors?). Since the authors mention that the both SpxB and LctO are part of a pathway that is responsible for the bulk of the ATP produced. The loss of these genes is likely to affect growth to a considerable degree, resulting in different bacterial loads between the WT and mutant conditions. Whether the results observed are majorly affected by these differences is presently impossible to determine, while it could be a very important factor. Different growth dynamics, might also result in a different ratio of live:dead bacteria at the time of measurement, which could release all sorts of molecules into the growth medium which could also affect the results. CFU/DNA copy numbers, or the addition of growth curves for the used strains (as supplementary material) would greatly improve the interpretation of the shown data. Intracellular growth might be quite different and difficult to measure, but in the data as presented, it is unclear what the dynamics of bacterial growth are. The observed difference in cytotoxicity could be simply because of differences in the bacterial load.

Minor comments

Many experiments are performed with 10 μ M Hb-Fe³⁺, which is shown to oxidize and result in cytotoxicity. However, most Hb *in vivo* would be Hb-Fe²⁺ to start with, which would first need to be oxidized to Hb-Fe³⁺ (which has been observed and reported previously). Hb-Fe³⁺ could then be oxidized to Hb-Fe⁴⁺. In reference 19, most Hb-Fe²⁺ is stated to be oxidized into Hb-Fe³⁺ in experimental conditions. It is unclear to me how much of the Hb-Fe²⁺ would be expected to be oxidized twice. So whether this oxidation of Hb-Fe³⁺ that is shown, would occur in physiological conditions. On what do the authors base the idea that Hb-Fe³⁺ oxidation could be expected in invasive disease and would it not be more logical to add Hb-Fe²⁺, which, if it also occurs in *in vivo* conditions, could be oxidized twice and also form the radical as observed?

Line 78: The authors mention Spx and LctO as the only enzymes that produce H_2O_2 , with percentages shown in line 82-83. There is at least one more enzyme, and possibly more, that also produce H_2O_2 : GlpO, involved in the glycerol metabolic process. GlpO catalyzes the following reaction: $O_2 + sn\text{-glycerol 3-phosphate} = \text{dihydroxyacetone phosphate} + H_2O_2$. As the mutants show, this likely does not contribute much to the overall H_2O_2 production by Spn, so it has little consequence for the content for the paper, but the statement that these two enzyme are solely responsible for H_2O_2 production is incorrect.

Line 158: Choose either using or with

Line 182: Washed once and then collected? Or just a single addition of PBS? Probably the latter, but the word washed confused me.

Line 191: Why was this concentration of Hb used? Does this have any physiological basis? It is used throughout the paper. Although it is possibly difficult to determine the level of Hb the pneumococcus is exposed to, it is important to understand the relevance of this amount of Hb in the *in vivo* situation.

Line 224-226: where the nuclei counted manually or using software algorithms?

Line 239: Why was the 100 μM H_2O_2 concentration used? Especially since you mention in line 379-380 that the TIGR4 strain used produces 700 μM within 6h of incubation.

Line 383-384: What was the conclusion?

Line 418: 8h for TIGR4 but 6h for EF3030

Line 434-435: H_2O_2 itself is already a ROS. It increases other intracellular ROS?

Line 439-440: Does it not just visualize an increase of intracellular H_2O_2 ? That the mutants do not produce? Have you tested the same experiment with just extracellular H_2O_2 in similar concentration to see whether the observed ROS is not just the increased H_2O_2 ? H_2O_2 alone did not affect tubulin, suggesting another actor, but a control for H_2O_2 would have been more correct.

Line 442-444: Have you quantified this observation?

Line 446-447: If H_2O_2 is the cause of cytotoxicity, why did the 100 μM H_2O_2 not affect cytoskeleton, why were no other concentrations tested to see if this is the result of just H_2O_2 or if other factors are involved? Should the oxidation of cell molecules also not be induced by the extracellularly added 100 μM H_2O_2 .

Line 459-460: This is worth showing

Line 481: Or that the amount of H_2O_2 produced (+/- 200 μM , fig. 3E) was sufficient for the cytotoxicity seen, especially since the cytotoxicity dropped from ~25% in figure 1B in 6 hours to 7% in 14 hours (3A)? Perhaps this experiment is only showing that Hb-Fe³⁺ can reduce H_2O_2 .

Line 485: It is also possible the cells reduced the produced H_2O_2 , although increased catalase would have been expected in the TIGR4 16h condition in fig 1G.

Line 516: You show that growth is significantly affected in biofilm in 5D, and although not significant, over a 100-fold lower CFU in the HU condition is quite a lot less.

Line 535: Typo of *S. pneumoniae*

Line 543-544: Would you not expect more of a signal in the mutants than as well? There is some in the TIGR4 double mutant, but nothing in the EF30303 mutant.

Line 586-587: It does not necessarily show that the DNA damage that is observed is part of the cause of the cytotoxicity. It would also be interesting to add a bacteria but no Hb-Fe³⁺, as well as a double mutant, as a control to see whether the damage is caused by oxidized components, H_2O_2 or the bacteria itself. Was the H_2O_2 concentration measured in the growth of R6 with Hb-Fe³⁺? Otherwise the DNA damage could still be a result of H_2O_2 .

Line 593: Again, what is this concentration based off?

Figure 1:

Showing the results of statistical comparison between two conditions in a graph seems to be chosen somewhat randomly, as in 1I for example.

1C: Were CFU counts/ODs measured to determine if the bacterial load was similar? Otherwise the effect could be due to the bacterial growth being affected by the catalase. If it is cytotoxic for A549 cells in those concentrations, perhaps it is the same for the pneumococci.

1D/1H: Why were two different timepoints chosen for the different strains? Additionally, there seem to be no bacteria in the TIGR4 DAPI stain?

1G: Would an effect of H_2O_2 on tubulin not be expected if that is what is the main difference between WT and the double mutant? The expected concentration is different though than the ~700 μM that was mentioned previously. The authors do not make note of this observation, or try to explain what is seen.

Figure 2: Legend repeats part of Fig. 1 legend.

Response to Reviewers

Dear Dr. Navarro-Garcia,

Thank you for the providing the opportunity for us to submit a revised version of my manuscript titled "Oxidation of hemoproteins by *Streptococcus pneumoniae* collapses the cell cytoskeleton and disrupts mitochondrial respiration leading to cytotoxicity of human lung cells". We have greatly appreciated the feedback and are thankful for the reviewer's valuable comments on the manuscript. Based on their suggestions, the manuscript has undergone revision and the changes within the revised submission have been highlighted.

Below are the point-by-point responses to each reviewers' comments.

Comments from Reviewer 1

- 1. The length of the manuscript is very long and suggested to shorten the introduction and materials and methods sections.**

Response: Thank you for the suggestion, the introduction and materials and methods sections have been shortened where applicable. Due to comments from Reviewer 2, the manuscript has lengthened slightly with suggested experiments that have now been included. We hope that these additions are amenable despite the minimal increase in length.

- 2. The authors have used already published methods to validate their hypothesis and it is suggested to use the references instead of providing detailed methodology.**

Response: We agree, certain methods in this section have now been referenced and certain detailed descriptions removed.

- 3. Figure numbers are missing in each page in the bottom of the manuscript. It was difficult to correlate/determine the figures in the text.**

Response: We apologize for the confusion; after checking the manuscript as well as the figures alone, figure numbers can now be found in the resubmission documents.

Comments from Reviewer 2

- 1. (Major Comment) There is no data shown or mentioned regarding the differences in growth between strains, nor is the amount of bacteria present at most measurements known to the reader....The observed difference in cytotoxicity could be simply because of differences in the bacterial load.**

Response: Thank you so much for this response, we agree that that the addition of growth curves/CFU counts would improve interpretation of the data. We have performed these growth curves and have included them in the revised manuscript. As seen in the revised Figure 1A, growth of both TIGR4 $\Delta spxB\Delta lctO$ and EF3030 $\Delta spxB\Delta lctO$ is significantly higher than their WT counterparts. This signifies that despite the increase in growth of the $\Delta spxB\Delta lctO$ strains, there is still minimal cytotoxicity indicating cytotoxicity seen in manuscript Fig. 1B and 1C is due to Spn-H₂O₂ alone and not an effect of the strain's growth or bacterial load. As well, the measure of cytotoxicity is done through the quantification of lactate dehydrogenase (LDH) and any molecules released during bacterial growth are

presumed to not have an effect on measurements. While Spn does produce a homolog of LDH, it is below the level of detection in the kit used so the only measurable LDH seen is from the human alveolar A549 cells (1). In reference to CFU/mL, those experiments for catalase have been performed and response to that can be seen in comment #24.

- 2. Many experiments are performed with 10 μ M Hb-Fe³⁺, which is shown to oxidize and results in cytotoxicity...On what do the authors base the idea that Hb-Fe³⁺ oxidation could be expected in invasive disease and would it not be more logical to add Hb-Fe²⁺, which, if it also occurs in in vivo conditions, could be oxidized twice and also from the radical as observed.**

Response: The general oxidation of hemoglobin with H₂O₂ in blood begins with Hb-Fe²⁺ to Hb-Fe⁴⁺ and then auto reduction of Hb-Fe⁴⁺ to Hb-Fe³⁺ (2). It is not uncommon for Hb-Fe³⁺ to be oxidized again to create Hb-Fe⁴⁺. These oxidations would also include the toxic radical intermediates between each state. We appreciate the suggestion of beginning the experiments with Hb-Fe²⁺ as there would be a higher presence of Hb-Fe²⁺ in blood during infection but there would still be initial oxidation of Hb-Fe²⁺ to Hb-Fe³⁺ and further to Hb-Fe⁴⁺. We choose to start these reactions at Hb-Fe³⁺, downstream of initial oxidation as we have previously observed that there is no differences in downstream oxidation when starting with Hb-Fe³⁺ rather than Hb-Fe²⁺ (3). As well, the process of introducing hemoglobin into experiments is an easier process with Hb-Fe³⁺ than Hb-Fe²⁺. If Hb-Fe²⁺ were used, oxidation to Hb-Fe⁴⁺ would still be seen and similar radical formation be observed.

- 3. Line 78: The authors mention SpxB and LctO as the enzymes that produce H₂O₂, with percentages shown in line 82-83... As the mutants show, this likely does not contribute much to the overall H₂O₂ production by Spn, so it has little consequence for the content in the paper, but the statement that these two enzymes are solely responsible for H₂O₂ production is incorrect.**

Response: Thank you for pointing this out, it has been edited so SpxB and LctO aren't regarded as the sole producers of H₂O₂ for Spn (Lines 76-80).

- 4. Line 158: Choose either using or with.**

Response: Have changed to "using".

- 5. Line 182: Washed once and then collected? Or just a single addition of PBS? Probably the latter, but the word washed confused me.**

Response: We apologize for the confusion. In this method of collecting bacteria for inoculum, sterile PBS is pipetted on an incubated Blood Agar Plate (BAP) containing bacterial growth. The PBS is then repeatedly pipetted up and down over the BAP to collect bacteria in a suspension of PBS (inoculum), this is what we term "washing". This inoculum is then used for subsequent experiments. For less confusion to other readers, "washing" has been changed to "collected".

- 6. Line 191: Why was this concentration of Hb used? Does this have any physiological basis? It is used throughout the paper. Although it is possibly difficult to determine the level of Hb the pneumococcus is exposed to, it is important to understand the relevance of this amount of Hb in the *in vivo* situation**

Response: While this concentration does not match the exact physiological hemoglobin concentration, where levels can vary, hemoglobin Soret peaks at ~415 nm and alpha and beta chain peaks at ~540 and

~570 respectively have been found within our lab to be measurable at a 10 μ M concentration during a reasonable time course study. Other reported and standardized concentrations have previously been 20 μ M (4, 5), though full oxidation of this concentration by Spn-H₂O₂ would be longer than the 6 h that is observed with the 10 μ M concentration. This is the *in vitro* proof of concept of what would occur *in vivo*.

7. Line 224-226: [Were] the nuclei counted manually or using software algorithms?

Response: These nuclei were counted using software algorithms with the NIS-Elements Basic Research Software, version 4.30.01 build 1021.

8. Line 239: Why was the 100 μ M H₂O₂ concentration used?

Response: We agree with your point here and thank you for pointing this out. This experiment has been performed again with a similar concentration to what Spn WT strains would produce and the updated figure and results which can be found within the manuscript. A549 cells that were treated with 750 μ M of H₂O₂ still show no change in tubulin signal. While it would be hypothesized that the addition of a similar level of H₂O₂ to that of Spn-H₂O₂ production would cause loss, it is important to note that over a 16 h time course that is presented here, Spn produces H₂O₂ continuously. The H₂O₂ addition at a similar level is done once at the beginning of the time course and acts as a limiting reagent to be oxidized by the cells where cells can then recover after the H₂O₂ is no longer present.

9. Line 383-384: What was the conclusion?

Response: The conclusion for cytotoxicity results can be found in the following paragraph. This line here is to introduce the reason for using a Δ *spxB* Δ *ctO* mutant rather than a single mutant in either enzyme, which is that both single mutants are cytotoxic to other bacteria when incubated alongside Spn.

10. Line 418: 8h for TIGR4 but 6h for EF3030

Response: While EF3030 exhibits lower density and growth than TIGR4 as shown in the newly added growth curves, EF3030 produces higher amounts of H₂O₂ than TIGR4 within the same time period. Due to EF3030's increased production of H₂O₂, and to show a similar level of cytoskeleton and tubulin loss as TIGR4, this occurs at 6h rather than 8h. This phenomenon has not yet been addressed but would be an interesting future study to perform.

11. Line 434-435: H₂O₂ itself is already a ROS. It increases other intracellular ROS?

Response: The increase in ROS that H₂O₂ produced by Spn would cause is our hypothesis. This is a proof of concept that Spn is able to produce H₂O₂ intracellularly after invasion of cells and/or that the H₂O₂ produced by Spn extracellularly are able to diffuse into cells.

12. Line 439-440: Does it not just visualize an increase in intracellular ROS? That the mutants do not produce? Have you tested the same experiment with just extracellular H₂O₂ in similar concentration to see whether the observed ROS is not just the increased H₂O₂? H₂O₂ alone did not affect tubulin, suggesting another actor, but a control for H₂O₂ would have been more correct.

Response: This kit (CellRox Green Reagent, for oxidative stress detection, Invitrogen) detects all ROS present within the cell, including H₂O₂. We agree that this experiment could include a H₂O₂ control and within the revised manuscript, there now includes a H₂O₂ control with similar concentrations that Spn would produce. With a similar level of H₂O₂ (750 μM), there is a small increase in ROS presence but no significant difference in measurable ROS. Reasoning for this mimic what most likely occurs in the Western blot experiments and explanation can be found in response to comment #8.

13. Line 442-444: Have you quantified this observation?

Response: Thank you for this suggestion, we have not quantified this observation. Though it would provide supplemental data to what is currently in Fig 2C, we believe that visual representation of increase in ROS represented by an increase in green fluorescence is agreeable to its conclusion.

14. If H₂O₂ is the cause of cytotoxicity, why did the 100 μM H₂O₂ not affect cytoskeleton, why were no other concentrations tested to see if this is the result of just H₂O₂ or if other factors are involved? Should the oxidation of cell molecules also not be induced by the extracellularly added 100 μM H₂O₂.

Response: We agree with your statement here that H₂O₂ would show an effect. We believe you are referring to the Western blot figure in Figure 1. At the 100μM concentration, there is no loss of tubulin cytoskeleton, but this does not reflect the amount of H₂O₂ Spn is capable of producing. Experiments have since been performed with other H₂O₂ concentrations and have reported the 750μM concentration as this is most similar to the concentration that Spn would produce. A new figure 1H reflects new data with this concentration. Explanation for what is seen in this new figure can be seen in the manuscript as well as in response to comment 8. Due to the response, we have seen at 100μM, A549 cells seem to be hardy against lower levels of H₂O₂ added once at the beginning of experiments. As well, Spn is a continuous producer of H₂O₂ capable of causing continuous repetitive damage to cells.

15. Line 459-460: This is worth showing

Response: Thank you for the suggestion here. The oxidation that is seen during the initial steps of the experiments exhibits similar oxidation curves as what is seen in Fig. 3D, 4A, and 4C. We believe the manuscript provides ample data to support our hypothesis and therefore, we have decided to not include this data within this section of results.

16. Line 481: Or that the amount of H₂O₂ produced (+/- 200μM, fig. 3E) was sufficient for the cytotoxicity seen... Perhaps this experiment is only showing that Hb-Fe³⁺ can reduce H₂O₂.

Response: This experiment is to observe, after Hb-Fe³⁺ oxidation by Spn-H₂O₂, would cause cytotoxicity. There is the possibility of residual H₂O₂ causing cytotoxicity though in conjunction with the findings that a tyrosyl radical is formed through Hb-Fe³⁺ oxidation by Spn-H₂O₂, it is plausible the radical is causing cytotoxicity.

17. Line 485: It is also possible the cells reduced the produced H₂O₂, although increase in catalase would have been expected in the TIGR4 16h condition in fig 1G.

Response: Yes, this experiment here is addressing that the Spn-H₂O₂ is reacting with targets within the cells and/or the cells themselves in a similar reaction with what is seen with Spn-H₂O₂ and Hb-Fe³⁺, as

there is a decrease in measurable H₂O₂. This would indicate that there is reduction of H₂O₂ by cells or targets within cells to create radicals leading to cytotoxicity.

18. Line 516: You show that growth is significantly affected in biofilm in 5D, and although not significant, over a 100-fold lower CFU in the HU conditions is quite a lot less.

Response: We agree with your statement here. While the data does appear to be visually significant, due to a n=2, any statistical analysis being performed is reporting no significance.

19. Line 535: Typo of *S. pneumoniae*

Response: Thank you for pointing this out, this typo has been edited to the correct spelling of *S. pneumoniae*.

20. Line 543-544: Would you not expect more of a signal in the mutants [then] as well? There is some in the TIGR4 double mutant, but nothing in the EF3030 mutant.

Response: We agree, this would be expected given the control. As the control in this figure is THY + Hb-Fe³⁺, the THY media (Todd-Hewitt Broth that contains 0.5% yeast extract) alone contains redox active compounds/transition metal ions able to catalyze minimal radical formation in aerobic conditions. These compounds that may catalyze this reaction with DEPMPO are the beef heart infusion, which contains undigested proteins and metal ions, that is found in THY and combination of proteins and peptides in yeast extract. The reasoning why this minimal signal would not be seen in the $\Delta spxB\Delta ctO$ mutants is most likely due to the consumption of these compounds by Spn.

21. Line 586-587: It does not necessarily show that the DNA damage that is observed is part of the cause of the cytotoxicity. It would be interesting to add bacteria but no Hb-Fe³⁺, as well as a double mutant, as a control to see whether the damage is caused by oxidized components, H₂O₂ or the bacteria itself. Was the H₂O₂ concentration measured in the growth of R6 with Hb-Fe³⁺? Otherwise the DNA damage could still be the result of H₂O₂

Response: Thank you for this comment, we have performed this suggested experiment. This data set overall (Fig. 6F) is referring to the capacity Hb-Fe³⁺ oxidation by Spn-H₂O₂ has on DNA degradation. If there is capacity to degrade bacterial DNA, there is also capacity to degrade alveolar DNA, which is seen in Fig. 6E. Also, adding in a control of Spn+DNA would remove any questions of if DNA damage is caused by the bacteria itself. In the updated figure 6F and within the results section, there is no DNA loss when R6 Spn is inoculated with TIGR4 DNA. As there is no significant difference seen in loss of DNA when R6 Spn is introduced, we would also expect there to be no difference as well with a $\Delta spxB\Delta ctO$ strain, so it was not included in the experiment. During the time of this experiment, H₂O₂ was not measured but R6 is known to produce similar H₂O₂ levels to that of TIGR4.

22. Line 593: Again, what is this concentration based off?

Response: This concentration of 56μM is based off of quantifiable Soret and alpha and beta chain peaks pre and post oxidation by Spn-H₂O₂ when measured on the spectrophotometer. Lower concentrations of Cytochrome *c* do not exhibit quantifiable/measurable peaks where we are able to see the oxidation phenomenon.

23. Figure 1: Showing the results of statistical comparison between two conditions in graph seems to be chosen somewhat randomly, as in 1I for examples.

Response: We agree, the statistical comparisons have been reformatted so that the comparisons being made are clear to the reader and aren't chosen randomly, thank you.

24. Figure 1: 1C: Were CFU counts/ODs measured to determine if the bacterial load was similar? Otherwise the effect could be due to the bacterial growth being affected by catalase. If it is cytotoxic for A549 cells in those concentrations, perhaps it is the same for pneumococci.

Response: CFU counts/OD's were not measured at the time of this experiment, but we agree with your statement here. If catalase was cytotoxic to A549 cells, there is a possibility that it is also cytotoxic to Spn. To test this and to supplement the data in 1C, TIGR4 was cultured with 400U and 1000U catalase. Notably, there is significant increase in the growth found when TIGR4 is grown in the presence of either 400U or 1000U compared to control. This signifies that catalase is not cytotoxic to Spn and does not affect its growth or ability to cause cytotoxicity and the phenomenon observed here is due to Spn- H_2O_2 's inability to cause cytotoxicity as it is being scavenged by catalase. A new figure incorporating this data is reflected in the revised manuscript.

25. Figure 1: 1D/1H: Why were two different timepoints chosen for the different strains? Additionally, there seems to be no bacteria in the TIGR4 DAPI strain?

Response: For the differing in timepoints between the strains, please see response under comment 10. We also apologize for the oversight in the addition of the wrong confocal image, we appreciate you catching this. This image has been replaced with the correct version with a representative image showing bacterial presence in the DAPI stain.

26. Figure 1: 1G: Would an effect of H_2O_2 on tubulin not be expected if that is what is the main difference between WT and the double mutant? The expected concentration is different though than the $\sim 700\mu M$ that was mentioned previously. The authors do not make note of this observation or try to explain what is seen.

Response: We agree with your statement here, this concern has been addressed, and the response can be found in both comment #8 and #14.

27. Figure 2: Legend repeats part of Fig. 1 legend.

Response: Thank you for noting this, the Figure 2 legend has been fixed and the edited version of the is reflected in the revised manuscript.

References

1. Kahlert CR, Nigg S, Onder L, Dijkman R, Diener L, Vidal AGJ, Rodriguez R, Vernazza P, Thiel V, Vidal JE, Albrich WC. 2023. The quorum sensing com system regulates pneumococcal colonisation and invasive disease in a pseudo-stratified airway tissue model. *Microbiol Res* 268:127297.
2. Alayash AI. 2022. Hemoglobin Oxidation Reactions in Stored Blood. *Antioxidants (Basel)* 11.
3. Alibayov B, Scasny A, Khan F, Creel A, Smith P, Vidal AGJ, Fitisemanu FM, Padilla-Benavides T, Weiser JN, Vidal JE. 2022. Oxidative Reactions Catalyzed by Hydrogen Peroxide Produced by *Streptococcus pneumoniae* and Other Streptococci Cause the Release and Degradation of Heme from Hemoglobin. *Infect Immun* 90:e0047122.
4. Akhter F, Womack E, Vidal JE, Le Breton Y, McIver KS, Pawar S, Eichenbaum Z. 2020. Hemoglobin stimulates vigorous growth of *Streptococcus pneumoniae* and shapes the pathogen's global transcriptome. *Sci Rep* 10:15202.
5. Akhter F, Womack E, Vidal JE, Le Breton Y, McIver KS, Pawar S, Eichenbaum Z. 2021. Hemoglobin Induces Early and Robust Biofilm Development in *Streptococcus pneumoniae* by a Pathway That Involves comC but Not the Cognate comDE Two-Component System. *Infect Immun* 89.

Re: Spectrum02912-23R1 (Oxidation of hemoproteins by *Streptococcus pneumoniae* collapses the cell cytoskeleton and disrupts mitochondrial respiration leading to cytotoxicity of human lung cells)

Dear Dr. Jorge E. Vidal:

Your manuscript has been accepted, and I am forwarding it to the ASM production staff for publication. Your paper will first be checked to make sure all elements meet the technical requirements. ASM staff will contact you if anything needs to be revised before copyediting and production can begin. Otherwise, you will be notified when your proofs are ready to be viewed.

Sincerely,
Fernando Navarro-Garcia
Editor
Microbiology Spectrum

Reviewer #2 (Comments for the Author):

The authors have sufficiently addressed my concerns